# Test-Time Canonicalization by Foundation Models for Robust Perception

**Utkarsh Singhal** [* 1]   **Ryan Feng** [* 2]   **Stella X. Yu** [1 2]   **Atul Prakash** [2]

## Abstract

Perception in the real world requires robustness to diverse viewing conditions. Existing approaches often rely on specialized architectures or training with predefined data augmentations, limiting adaptability. Taking inspiration from mental rotation in human vision, we propose FOCAL, a test-time robustness framework that transforms the input into the most typical view. At inference-time, FOCAL explores a set of transformed images and chooses the one with the highest likelihood under foundation model priors. This test-time optimization boosts robustness while requiring no retraining or architectural changes. Applied to models like CLIP and SAM, it significantly boosts robustness across a wide range of transformations, including 2D and 3D rotations, contrast and lighting shifts, and day-night changes. We also explore potential applications in active vision. By reframing invariance as a test-time optimization problem, FOCAL offers a general and scalable approach to robustness. Our code is available at: https://github.com/sutkarsh/focal.

## 1. Introduction

A robot navigating a cluttered home or a self-driving car on a street must reliably recognize objects despite changes in viewpoint, orientation, and lighting. Such robustness is essential for embodied perception in dynamic environments.

Human vision has evolved to excel at this task: We readily recognize familiar objects from diverse viewpoints while mentally rotating unfamiliar objects to typical views to facilitate recognition (Shepard & Metzler, 1971). Through natural visual exposure, we develop invariances that adapt to each object and context, generalizing to novel ones while preserving necessary distinctions. That is, human perception develops invariances shaped by ecological demands.

---
*Equal contribution   [1]UC Berkeley [2]University of Michigan. Correspondence to: Utkarsh Singhal <s.utkarsh@berkeley.edu>.

*Proceedings of the $42^{nd}$ International Conference on Machine Learning*, Vancouver, Canada. PMLR 267, 2025. Copyright 2025 by the author(s).

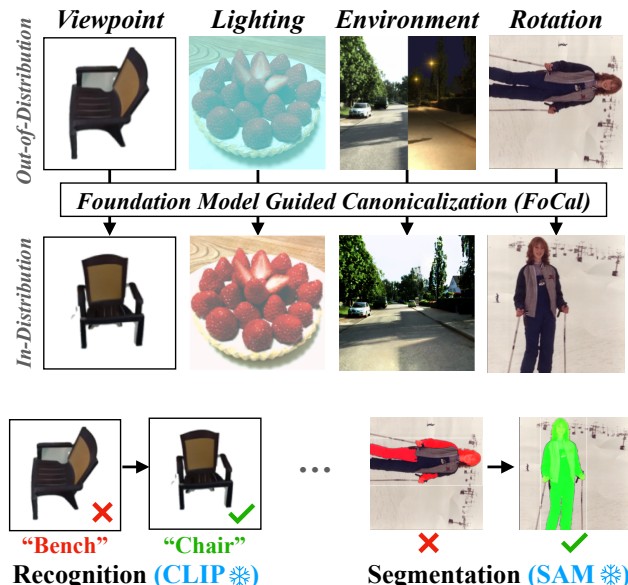

*Figure 1.* **Introducing FOCAL**: a test-time framework for approximate invariance to complex transformations at scale. **(top)** Given variations in viewpoint, illumination, environment, and rotation, FOCAL use foundation model priors to select a visually-typical version of the input. **(bottom)** out-of-distribution inputs can lead to incorrect predictions on CLIP ("bench") and incorrect segmentations on box-prompted SAM (missing body parts in the segmentation mask). Using FOCAL canonicalized versions fixes these errors. Thus, FOCAL offers a scalable, data-driven method for robust perception across diverse transforms.

In contrast, even leading foundation models exhibit brittleness (Figure 1): CLIP (Radford et al., 2021) misclassifies images from uncommon viewpoints, and SAM (Kirillov et al., 2023) incorrectly segments objects presented sideways. This vulnerability stems partly from the overrepresentation of upright poses and ideal lighting in internet training data – a phenomenon known as *photographer's bias*. Consequently, these models are unprepared for diverse visual inputs seen in embodied settings (Madan et al., 2024).

Data augmentation (DA) and equivariant networks are two common approaches to improve neural network robustness to *specific* transformations. DA exposes models to carefully selected transformations during training (e.g., rotations, crops), but struggles with rare classes (Zhou et al., 2022) and hurts some classes through over-regularization (Xiao et al.,

2021). Equivariant networks (Cohen & Welling, 2016) build mathematical symmetries (e.g., 2D rotations) directly into the architecture, but this approach does not extend to complex real-world transformations like 3D viewpoint changes.

Critically, both approaches *hard-wire* invariances into the model during training through either curated training data or the model architecture. This strategy is inherently rigid: it assumes prior knowledge of relevant transformations and often fails under distribution shifts such as novel viewpoints or lighting conditions beyond those seen during training.

We propose a fundamentally different approach: Instead of hard-wiring invariance during training, our **Fo**undation-model guided **Ca**nonicalization (FoCAL) achieves invariance by *reasoning over transformations at inference time* (Figure 1). Like human vision, which mentally rotates unfamiliar objects to canonical views for recognition (Shepard & Metzler, 1971), FoCAL leverages foundation models' visual priors to transform inputs toward *visually typical* views.

FoCAL uses a "vary and rank" strategy (Figure 2): It generates candidate transforms (e.g., via generative models) and selects the most visually typical instance by minimizing an energy function based on Stable Diffusion (SD) and CLIP. The optimized instance is then passed to downstream models for inference. This perspective connects to recent test-time scaling work where learned rankers select among outputs (Snell et al., 2024). However, FoCAL distinguishes itself by grounding ranking in data-driven visual typicality.

Our approach enables robust perception across diverse conditions, including changes in viewpoint, lighting, and environment, *without any additional training or architectural changes*. Invariance in our method does not arise from hardwired constraints, but is instead an emergent property of our test-time optimization. Unlike prior canonicalization work that requires transform-specific training (Mondal et al., 2023), FoCAL works for a wide range of datasets and transforms. By reframing invariance as *test-time optimization*, FoCAL offers a data-driven method for robustness at scale.

We evaluate FoCAL across a range of challenging transformations, including 3D-viewpoint, illumination, day-night changes, and 2D rotations. We find that FoCAL improves out-of-distribution performance of foundation models such as CLIP (Radford et al., 2021) on ImageNet (Deng et al., 2009) scale datasets. To our knowledge, this is a significant improvement in dataset size and transformation complexity compared to previous canonicalization work. Remarkably, FoCAL consistently matches or outperforms task-specific canonicalizers like PRLC (Mondal et al., 2023) *even in their trained settings*, despite operating entirely at test time.

**Contributions. (1)** We introduce FoCAL, a test-time, data-driven framework that leverages the visual priors of foundation models to achieve invariance. **(2)** We propose an

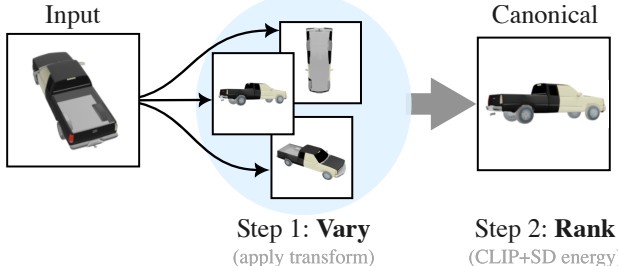

Input              Canonical

Step 1: **Vary**
(apply transform)

Step 2: **Rank**
(CLIP+SD energy)

*Figure 2.* **Foundation Model Guided Canonicalization**: Our method works in two steps: Vary and Rank. We first produce transformed variations of the input image and then rank them using energy functions derived from CLIP and Stable Diffusion. The minimizer of this energy function is the most visually typical version, which we refer to as the "canonical". This approach is training-free, transformation-agnostic, and highly generalizable.

approximate invariance method that scales to complex transformations, including 3D viewpoint shifts, lighting changes, and environmental variations. **(3)** We demonstrate the effectiveness of FoCAL through evaluations on modern models such as CLIP, OV-Seg, and SAM, across diverse datasets including ImageNet, COCO, Objaverse-LVIS, and CO3D.

Our results challenge the prevailing assumption that achieving invariance requires specialized training or architectural design, offering a path toward robust perception for embodied agents. By bridging insights from human mental rotation and modern test-time compute scaling (Zaremba et al., 2025), FoCAL demonstrates that test-time optimization can serve as a scalable approach to robust visual perception under real-world variation.

## 2. Robust Perception by Minimizing Foundation Model Energy Functions

We begin with a motivating example: a rotated image. When presented with an upside-down photograph, humans naturally mentally rotate it to understand its contents (Shepard & Metzler, 1971). How can we automatically find such "canonical" (or visually-typical) representations based on the priors in foundation models?

### 2.1. Problem setting

We denote the input image by $x \in \mathcal{X}$, where $\mathcal{X}$ is the space of images. Images may undergo transformations $t : \mathcal{X} \to \mathcal{X}$, such as rotations. We denote the set of such transformations (e.g., the group of image rotations) by $\mathcal{T}$.

Given an image, we may want to process it using an existing neural network (e.g., for recognition or segmentation). We denote this operation with a function $f : \mathcal{X} \to \mathcal{Y}$ with inputs $x \in \mathcal{X}$ and outputs $y \in \mathcal{Y}$, where $\mathcal{Y}$ is the space of outputs (e.g., classes or masks).

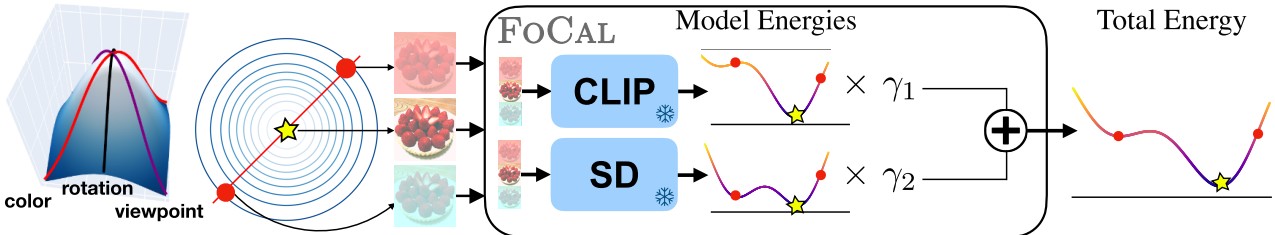

*Figure 3.* **Transformation distributions define a slice through the distribution of natural images, enabling us to use foundation models to canonicalize.** Given the complex distribution of natural data, which spans many transformations, we propose a common solution that applies across arbitrary transformation-based slices of the distribution (left). Given a particular slice (red curve in the example above), we simulate different versions of the input image along this slice of the distribution (e.g., color variations), using energy functions built on CLIP (Radford et al., 2021) and Stable Diffusion (Rombach et al., 2021) models for each sample (model energies in the figure above). We minimize the total energy to predict the canonical version (right, with the optimum shown by a star). In summary, our method uses internet-scale priors learned by foundation models to canonicalize diverse transformations.

For recognition, we want the function $\boldsymbol{f}$ to be *invariant* to the transformations in $\mathcal{T}$, meaning that it should produce the same output regardless of how the input is transformed. For example, if we rotate an image, the recognition function should still recognize the object in the image:

$$\text{Invariance:} \quad \boldsymbol{f}(t(\boldsymbol{x})) = \boldsymbol{f}(\boldsymbol{x}) \ \forall t \in \mathcal{T}$$

For segmentation, we want the function to be *equivariant* to the transformations in $\mathcal{T}$, meaning that the output should change predictably when the input is transformed. For example, if we rotate an image, the segmentation mask should also be rotated:

$$\text{Equivariance:} \quad \boldsymbol{f}(t(\boldsymbol{x})) = t'(\boldsymbol{f}(\boldsymbol{x})) \ \forall t \in \mathcal{T}$$

where $t'$ is the corresponding transformation in the output space $\mathcal{Y}$, like the segmentation mask rotating with the input.

We next explain how *canonicalization* achieves these desiderata in the context of recent work by Kaba et al. (2022) with simplified notation and more general transformations.

**Canonicalization**: Canonicalization refers to the process of transforming an input into a fixed "canonical" version regardless of the transformations applied. For rotations, this might look like uprighting an image so that it is always presented in a standard orientation.

Kaba et al. (2022) formalize this with a canonicalizer function $\boldsymbol{h} : \mathcal{X} \to \mathcal{T}$ that maps the input to the transformation needed to "canonicalize" it. The core idea is that the canonical image $\boldsymbol{h}(\boldsymbol{x})(\boldsymbol{x})$ is the same regardless of the input's transform, i.e., $\boldsymbol{h}(t(\boldsymbol{x}))(t(\boldsymbol{x})) = \boldsymbol{h}(\boldsymbol{x})(\boldsymbol{x})$ for all $t \in \mathcal{T}$.

Thus, for invariance, it is sufficient to apply the canonicalizer to the input before processing it with the function $\boldsymbol{f}$:

$$\text{let} \quad \tilde{\boldsymbol{x}} := t(\boldsymbol{x})$$
$$\boldsymbol{f}(\boldsymbol{h}(\boldsymbol{x})(\boldsymbol{x})) = \boldsymbol{f}(\boldsymbol{h}(\tilde{\boldsymbol{x}})(\tilde{\boldsymbol{x}}))$$

Here, $\boldsymbol{h}(\boldsymbol{x})$ effectively undoes the transformation on $\boldsymbol{x}$, achieving invariant output.

But how do we construct such a canonicalizer $\boldsymbol{h}$? Under mild conditions, Kaba et al. (2022) show that if the canonicalizer $h$ is defined as an energy minimizer over transformations, then it achieves the desired invariance and equivariance properties. Specifically, they show that the canonicalizer can be defined as:

$$h(\boldsymbol{x}) = \arg\min_{t \in \mathcal{T}} E(t(\boldsymbol{x})) \tag{1}$$

where $E : \mathcal{X} \to \mathbb{R}$ is a real-valued function referred to as the "energy function". Strikingly, this result does not require $E$ or $\boldsymbol{f}$ to be equivariant or invariant. Instead, the canonicalizer $\boldsymbol{h}$ emerges from the minimization process.

**Caveat:** This framework guarantees invariance for invertible transformations, but many important transformations are non-invertible (e.g., viewpoint shifts). However, in practice, we find that generative models can still provide *approximate* invariance for some easier cases (Figure 5).

Secondly, *invariant* doesn't imply *correct*: The chosen canonical can be out-of-distribution for the downstream model, leading to low accuracy. Mondal et al. (2023) attempt to fix this by training the canonicalizer on the target dataset. We instead pick the most visually typical (likely in-distribution) version with foundation model priors.

### 2.2. Key Insight: Slices of Natural Image Distribution

Building on this foundation, we make a crucial observation: an image $\boldsymbol{x}$ and all its transformed versions $t(\boldsymbol{x}) \ \forall t \in \mathcal{T}$ define a *slice* through the natural image distribution $p_{\text{data}}(\boldsymbol{x})$ (Figure 3). Within this slice, some versions appear more frequently in real-world data than others. We find the most likely version with visual priors in foundation models.

If we define an energy function $E \approx -\log p_{\text{data}}$, then minimizing $E$ over the transformed versions of $\boldsymbol{x}$ effectively finds the most probable version of the image within that slice. Denoting the 'best' transform $t^* = \arg\min_{t \in \mathcal{T}} E(t(\boldsymbol{x}))$,

Candidates      Energy Function

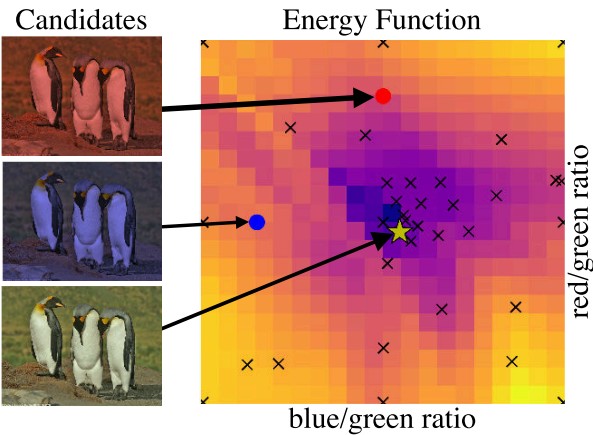

*Figure 4.* **Foundation Model Guided Canonicalization in continuous and multi-dimensional transform spaces**: Given an input image, we generate different transformed versions of the image (candidates). Each is ranked by a combined energy function as shown in Figure 3, with the minimum of this grid representing the canonical form. However, brute-force search is infeasible for continuous or multi-dimensional transform spaces such as color shift. Thus, we apply Bayesian Optimization for efficient optimization in such spaces. Black crosses show the locations of BO guesses, and the yellow star shows the location of the found optimum. BO finds the optimum without using hundreds of evaluations. Thus, combining energy functions and Bayesian optimization provides a general approach to canonicalization.

the canonical image $\boldsymbol{x}^* = t^*(\boldsymbol{x})$ is likely the most visually typical or informative version of the image. For example, if $\boldsymbol{x}$ is a rotated image, the canonical form $\boldsymbol{x}^*$ is likely the upright version of that image. For a color-shifted image, the canonical form is likely the version in typical lighting.

Since modern foundation models like CLIP and Stable Diffusion are trained on vast datasets of natural images, they implicitly learn visual priors about natural data. We can therefore extract energy functions $E \approx -\log p_{\text{data}}$ from these models to guide canonicalization.

Note that in our instance, the "canonicalized" views do not necessarily need to be exactly "upright" - the idea is simply that we try to map all inputs to the same, optimized view for invariance. As long as such a view is in-distribution, it is suitable for downstream recognizers and segmenters.

Next, we formalize energy functions and describe how to extract them from foundation models.

### 2.3. Energy Functions from Foundation Models

We extract energy functions from pre-trained foundation models, which can be readily combined with each other and with hand-designed priors.

**Energy-based models**: Any probability distribution $P_\theta(x)$

for a random variable $x \in \mathbb{R}^D$ can be written as:

$$P_\theta(x) = \frac{1}{Z(\theta)} e^{-E_\theta(x)}$$

where $Z(\theta)$ is the normalizing constant and $E_\theta : \mathbb{R}^D \to \mathbb{R}$ is the energy function. Small values of $E_\theta(x)$ correspond to more likely $x$.

**CLIP Energy**: Following Grathwohl et al. (2020), any classifier can be seen as an energy-based model. The classifier energy for $f_\theta$ with input $x$ and output $y$ is the negative logit:

$$E_\theta(x, y) = -f_\theta(x)[y]$$

For unconditional energy, Grathwohl et al. (2020) use log-sum-exp over all labels, but we simplify this using a combination of mean and max logits:

$$E_{\text{CLIP}}(\boldsymbol{x}; \alpha, \beta) = \left(\alpha \cdot \text{mean} - \beta \cdot \max_{c \in 1,2,\ldots,|C|}\right)\left(f_\theta(\boldsymbol{x})[c]\right)$$

where $\alpha, \beta \in \mathbb{R}$ are hyperparameters and $f_\theta(\boldsymbol{x})[c]$ is the class $c$ logit. We use CLIP ViT-H-14 (Ilharco et al., 2021) with cosine similarity of image & text embeddings as logits.

**Diffusion Energy**: Following Graikos et al. (2022), diffusion models provide effective image priors through free energy minimization:

$$E_{\text{diff}}(\boldsymbol{x}) = \frac{1}{T} \sum_{t=1}^{T} \mathbb{E}_{\boldsymbol{\epsilon} \sim \mathcal{N}(\mathbf{0}, \mathbf{I})}\left[\|\boldsymbol{\epsilon} - \boldsymbol{\epsilon}_\theta(\mathbf{x_t}, t)\|^2\right] \quad (2)$$

where $\mathbf{x_t} = \sqrt{\bar{\alpha}_t}\boldsymbol{x} + \sqrt{1 - \bar{\alpha}_t}\boldsymbol{\epsilon}$ is the noisy input, $\boldsymbol{\epsilon_\theta}$ is the pre-trained diffusion model, and $\bar{\alpha}_t = \prod_{i=1}^{t}(1 - \beta_i)$ with denoising schedule $\beta_t$. We use SD-2-base (Rombach et al., 2021), finding that $5 - 10$ steps are usually sufficient.

### 2.4. Foundation Model Guided Canonicalization

**'Vary' and 'Rank'**: Our method consists of two steps: *Vary*, where we produce variations of an input image, and *Rank*, where we rank the variations (Figure 2). We optimize the FoCaL energy function over transforms of the input $\boldsymbol{x}$:

$$t^* = \arg\min_{t \in \mathcal{T}} E_{\text{FoCaL}}\left(t(\boldsymbol{x})\right) \quad (3)$$

$$\boldsymbol{y} = f(t^*(\boldsymbol{x})) \quad (4)$$

where $\mathcal{T}$ is the set of transformations, $E_{\text{FoCaL}}$ is the energy function defined below, and $y$ is the invariant output.

**Combining energy functions**: We minimize the combined energy $E_{\text{FoCaL}}\left(t(\boldsymbol{x})\right)$ over all transformations $t \in \mathcal{T}$ to find the canonical version of the input image $\boldsymbol{x}$. This is done by solving the following optimization problem:

$$E_{\text{FoCaL}}\left(t(\boldsymbol{x})\right) = \gamma_1 E_{\text{CLIP}}(t(\boldsymbol{x})) + \gamma_2 E_{\text{diff}}(t(\boldsymbol{x})) \quad (5)$$

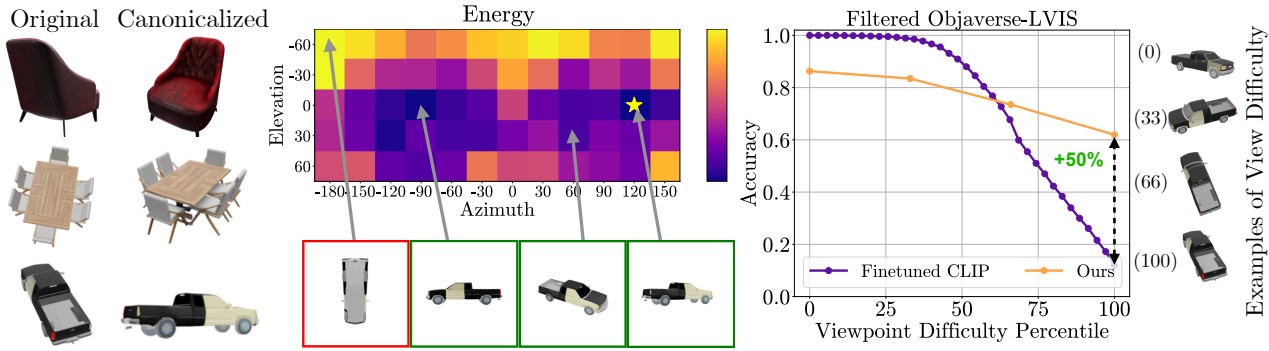

*Figure 5.* **Viewpoint canonicalization improves accuracy by improving viewpoint robustness: (Left)** Example renderings from Objaverse-LVIS before (original) and after (canonicalized) applying FOCAL. The canonicalized views provide more informative perspectives for object recognition. **(Middle)** Energy plot showing different elevations ($y$ axis) and azimuths ($x$ axis). Locations with lower energy (darker purple) correspond with more canonical views that can be more easily classified correctly (green box). In contrast, locations with high energy (yellow) tend to be incorrect (red box). Star denotes the optimal, selected view. **(Right)** Accuracy on quality-filtered Objaverse-LVIS across different viewpoints ranked by difficulty, with easier viewpoints on the left and harder viewpoints on the right. The finetuned CLIP model (OV-Seg) (purple line) exhibits a steep accuracy drop for difficult viewpoints, whereas FOCAL (orange line) maintains more stable performance, achieving 50% higher accuracy for difficult views. Examples on the right depict a truck render at difficulty percentiles ranging from 0 (easiest) to 100 (hardest). In summary, we find that FOCAL identifies more visually typical versions of input images, enabling higher performance on bad input 3D viewpoints.

where $\alpha, \beta, \gamma_1, \gamma_2 \in \mathbb{R}$ are hyperparameters.

Intuitively, CLIP energy focuses on semantics, selecting the image that most closely resembles a predefined category, while SD energy acts as a general appearance-based prior.

**Assumptions**: For invariance, we assume that the transformation is invertible. Under mild conditions used in Kaba et al. (2022), the energy optimization produces invariant outputs. For high downstream accuracy, we assume: (1) there is at least one in-distribution image in the set of transformed images, (2) the foundation models can be used as a prior where in-distribution images have lower energy than out-of-distribution ones, and (3) the downstream model performs best in-distribution. If the first two assumptions hold for a given sample, the energy minimization scheme returns an image that is in-distribution. If the third assumption holds, this process is likely to result in higher accuracy.

**Bayesian Optimization for Efficient Search**: While exhaustive search can be used to minimize the energy function in Section 2.4 for a small number of transformations (e.g., 8 rotations around the circle), it is infeasible for continuous and higher-dimensional transformations. This is critical since many common transformations are high-dimensional: color transformation (Figure 7) is 2D, "active vision" setting is 6D, and combining transforms is even higher-dimension.

Fortunately, this problem can be handled by Bayesian Optimization (BO) (Nogueira, 2014; Frazier, 2018) (Figure 4), a well-established method for efficient optimization in both low and mid-dimensional spaces. BO utilizes a probabilistic model, such as a Gaussian Process (GP), to approximate the objective function based on a few evaluations. We utilize Bayesian Optimization (BO) with a Gaussian Process (GP) using an RBF kernel and the Expected Improvement (EI) acquisition function (Jones et al., 1998) to balance exploration and exploitation. This approach balances exploring uncertain regions with exploiting promising areas, typically finding good solutions in 50-100 evaluations.

BO is commonly used for optimization problems like hyperparameter search and has been successfully applied to high-dimensional problems such as optimization in the latent space of generative models (Maus et al., 2022; Gómez-Bombarelli et al., 2018; Castro et al., 2022; Tripp et al., 2020). For even higher-dimensional problems, gradient-based optimization is another potential alternative. We leave the exploration of gradient-based methods to future work.

## 3. Experiments

This section outlines our experimental settings and results. We pick classification and segmentation as standard tasks and test FOCAL on diverse transformations (viewpoint shift, color, contrast, day-night, active vision). We find that FOCAL generalizes across many different settings and even beats PRLC (Mondal et al., 2023) on 2D rotations with their jointly-trained classifiers. These results show the generalizability and wide applicability of FOCAL as a canonicalizer.

### 3.1. 3D Viewpoints

We evaluate FOCAL as a method for approximate invariance to viewpoint transforms on Objaverse-LVIS (Deitke et al.,

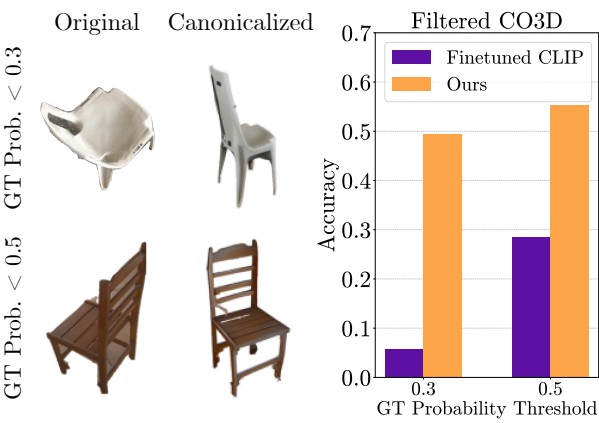

*Figure 6.* **Viewpoint canonicalization improves accuracy on CO3D.** For each CO3D video, we randomly sample a frame with a ground truth label probability below two different difficulty thresholds (0.3, 0.5). **(Left)** We show two examples on the left of an input image that is incorrectly classified at first but is correctly classified as a chair after running through FOCAL. **(Right)** Accuracy comparison. FOCAL (orange) outperforms finetuned CLIP (OV-Seg) (purple), highlighting its ability to (approximately) canonicalize viewpoints in realistic settings.

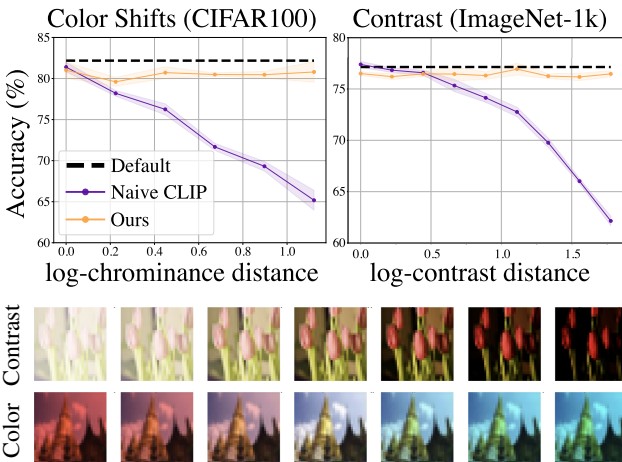

*Figure 7.* **FOCAL canonicalizes illumination (color & contrast)** We evaluate FOCAL using CLIP on two different datasets (CI-FAR100, ImageNet-1K) and transformations (color, contrast). **(a)** FOCAL improves CLIP's accuracy on color-shifted images by 9.9% on average and nearly 15% for larger shifts. **(b)** FOCAL improves CLIP's accuracy under contrast changes by 4.1% on average and nearly 12% for larger contrast shifts. These results highlight FOCAL's ability to handle lighting transformations.

2023) and CO3D (Reizenstein et al., 2021). Objaverse-LVIS dataset contains 46K 3D assets and CO3D contains 19K real multiview video sequences with object segmentations.

We begin by filtering datasets to ensure high-quality objects and viewpoints. For Objaverse-LVIS, we noticed cases of misleading and overlapping labels and thus filtered out such objects. For CO3D, we use sequences with sufficient variation in viewpoint quality. Given the filtered videos, we then sample one of the poor viewpoints (i.e., ground truth probability of $< t$ for some threshold $t$) and filter out images with blurriness or segmentation errors (Appendix B.1).

We use TRELLIS (Xiang et al., 2024) to produce viewpoint variations for each input. Specifically, we render views on the sphere at $30°$ intervals for the "vary" stage (Appendix B.1). We rank these renders using FOCAL energy and pick the view with the minimum energy. Since TRELLIS generates background-removed images, we use OV-Seg (Liang et al., 2023), a version of CLIP fine-tuned on background-removed images, as our classifier.

**Objaverse-LVIS:** We include example renders and accuracy plots over viewpoints in Figure 5. We find that FOCAL often canonicalizes input images from out-of-distribution views to canonical forms, increasing accuracy. We demonstrate this by ranking viewpoints by ground truth classification probability (best to worst) and computing average accuracy per rank. Figure 5 shows that FOCAL significantly improves classification accuracy for difficult viewpoints (12.0% to 62.0% on the worst viewpoints) and improves the overall stability (max accuracy - min accuracy).

This result shows the generalizability of FOCAL to 3D viewpoint transformations. To our knowledge, this result represents a significant step forward in invariance methods applied to viewpoint transformations. We include further comparisons against test-time augmentation (TTA) and interesting findings that suggest that TRELLIS (Xiang et al., 2024) itself can canonicalize to an extent on inputs near its training distribution in Appendix A.2.

**CO3D:** We then test FOCAL's ability to improve the accuracy of our selected CO3D frames. We show in Fig. 6 for both a threshold of $t = 0.3$ and a threshold of $t = 0.5$ that FOCAL improves the accuracy of our frames by 43.8% and 26.8%, respectively. We also show example renders and additional comparisons in the Appendix (Figure 11 and A.7). This result further shows our capability to canonicalize over 3D viewpoints on real-world images.

### 3.2. Lighting (color and contrast)

We evaluate chrominance (color) and contrast transformations on CIFAR100 (Krizhevsky et al., 2010) and ImageNet (Deng et al., 2009) with CLIP (Radford et al., 2021). Unlike classic methods, FOCAL achieves invariance without requiring specialized architectures trained on curated ground truth (Hernandez-Juarez et al., 2020).

As shown in Figure 7, FOCAL improves accuracy by 9.9% for color shifts and 4.1% for contrast shifts over vanilla CLIP, with gains reaching 15% and 12% for extreme transformations. While not surpassing supervised approaches

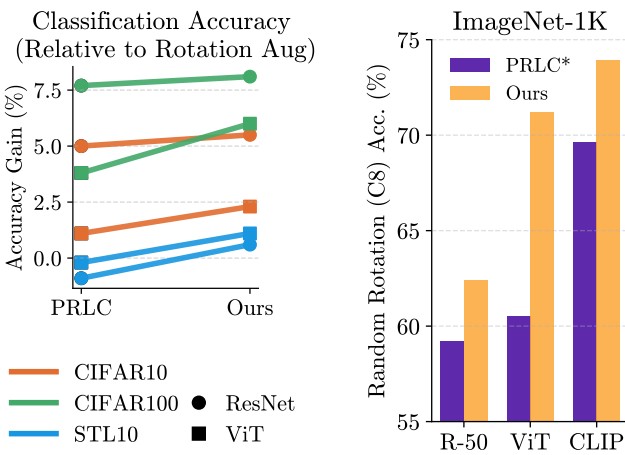

Figure 8. **FOCAL beats PRLC on their image classification evaluations**: We evaluate FOCAL on 2D rotations following the setting of Mondal et al. (2023). **(a)** Left plot shows classification accuracy gains relative to rotation augmentation baseline across multiple datasets (CIFAR10, CIFAR100, STL10) using both ResNet-50 and ViT architectures. Each paired line represents a different evaluation setting, with FOCAL (Ours) consistently matching or outperforming PRLC in each setting. **(b)** We evaluate each of PRLC's trained canonicalizers on an unseen dataset (ImageNet-1K) and pick the best canonicalizer for each architecture. We find that FOCAL still beats them as PRLC's canonicalizers struggle to generalize outside their training setting. In short, FOCAL outperforms trained canonicalizers of Mondal et al. (2023) on seen as well as unseen datasets, highlighting FOCAL's generalizability.

like (Barron & Tsai, 2017; Hernandez-Juarez et al., 2020) (Appendix A.9), FOCAL maintains stable accuracy across variations (details in Appendix B.2), demonstrating lighting canonicalization without specific training or architecture.

### 3.3. 2D Rotation (Comparison against PRLC)

**Classification:** We compare against PRLC (Mondal et al., 2023) on their 2D rotation settings ($C8$) using PRLC-trained ViT (Dosovitskiy et al., 2021) and PRLC-trained ResNet50 (He et al., 2016) models across CI-FAR10 (Krizhevsky et al., 2010), CIFAR100 (Krizhevsky et al., 2010), and STL10 (Coates et al., 2011).

We report accuracy on upright images, rotated images (on $C8$ rotations), pose accuracy (i.e., did the canonicalizer pick the correct $C8$ rotation), and pose error (i.e., the average error in degrees) (details in Appendix, Table 3, summary in Figure 8). We find that FOCAL matches or beats PRLC's specially trained canonicalizers on *every one of their evaluated settings* zero-shot (Figure 8).

We extend this setting further to CLIP and ImageNet (Deng et al., 2009), finding that FOCAL generalizes much better to this setting (by up to $+11\%$) than PRLC, which struggles to generalize beyond its training scope (Figure 8 and Table 5).

We also benchmark how our method helps in combination

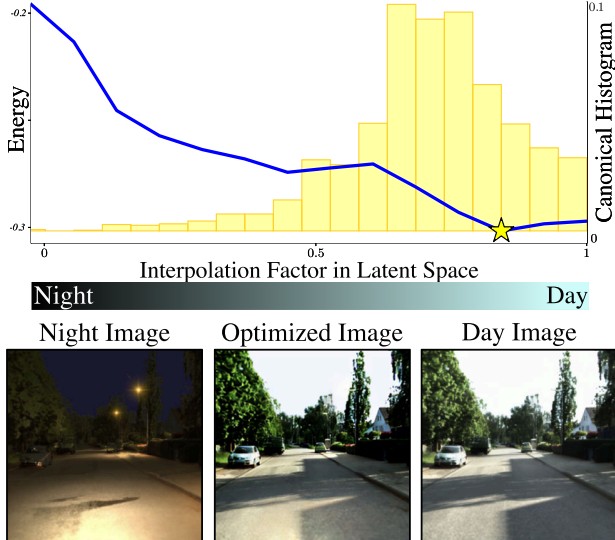

Figure 9. **FOCAL can canonicalize day-night transformations**. We evaluate the CLIP energy function as we interpolate between the day-night image pairs in the latent space of a diffusion model, finding that FOCAL prefers day images with $91\%$ accuracy. We also plot the energy function (purple) and the histogram (yellow) of the energy optimum for 5000 pairs, showing that the chosen images are close to daytime. Thus, FOCAL can choose more visually typical and informative daytime images in the day-night transformation setting in the latent space of a diffusion model.

with DA. We train a ResNet-32 on CIFAR10 for 100 epochs with LR $10^{-3}$. The C8 rotated accuracy is $72\%$ with just DA, $72.1\%$ with just FOCAL, and $73.2\%$ with FOCAL+DA. We leave a rigorous study of this setting to future work.

**Segmentation**: We also evaluate PRLC's segmentation setting with SAM (Kirillov et al., 2023) using the pre-trained SAM canonicalizer supplied by the authors. Here, we find that our method matches their pre-trained canonicalizer in mAP ($65.9$) while achieving $+2.1\%$ better pose accuracy.

These results show that FOCAL not only matches PRLC's performance on their trained datasets without *requiring any training* but also outperforms PRLC in novel settings beyond their original training scope. Further cross-dataset and cross-classifier generalization results are in Appendix A.

### 3.4. Day-Night transformation

We apply FOCAL on day-night transformations modeled by UrbanIR (Lin et al., 2024), a state-of-the-art relighting model. We use their pre-trained KITTI (Geiger et al., 2012) checkpoints and relighting code to render 2000 pairs of day-night frames. Since it is unclear what classes should be used for CLIP energy, we simply use 1 single class "street." This energy picks day images $91\%$ of the time.

To further study FOCAL's energy function, we create more variations of each scene that lie *between* day and night im-

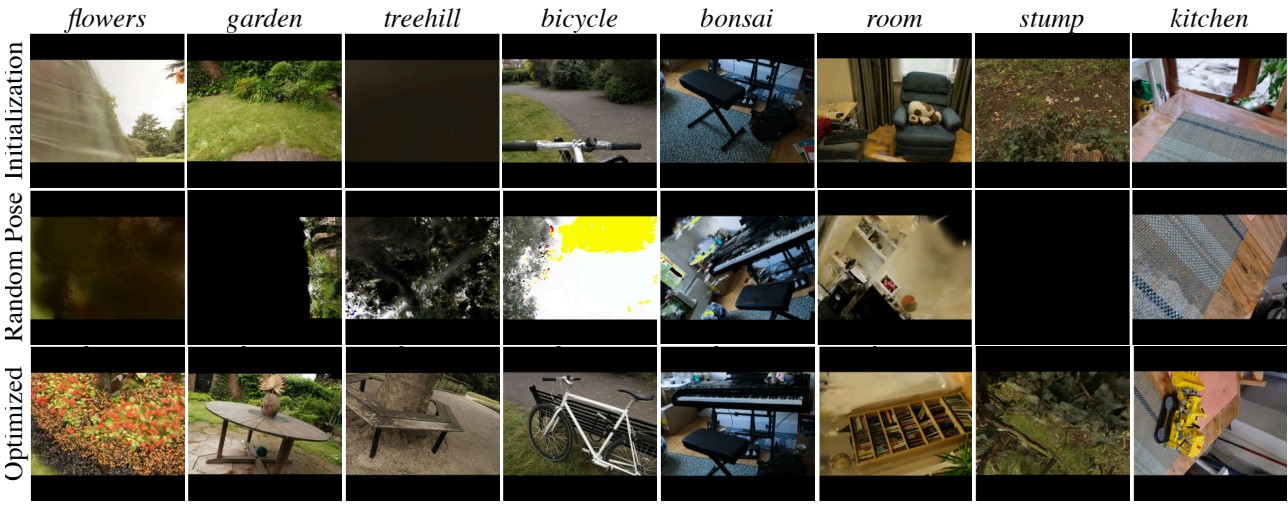

*Figure 10.* **Applying FOCAL to active vision**: We optimize a 6-DoF camera pose (x, y, z, yaw, pitch, roll) in a 3D virtual environment. We use Bayesian optimization to minimize FOCAL energy, and plot the results above along with randomly chosen poses for comparison. The first row shows the initialization, second row shows a random pose, and the third row shows the result from FOCAL. We find that the camera naturally focuses on salient objects and tends to maintain upright angles, reflecting behaviors observed in simpler settings (green underline). Although some failure cases remain (red underline), these results illustrate FOCAL 's ability to generalize to significantly more complex transformations and viewing conditions in a zero-shot manner.

ages by interpolating their latent vectors in the latent space of Stable Diffusion 2 (Rombach et al., 2021). This process creates a restricted latent space containing variations of the scene. We plot the energy function over this space (Figure 9). We then use FOCAL to canonicalize this transform by optimizing the energy function in the latent space, finding that the canonicals concentrate around day images.

### 3.5. Active vision

We also show an exploratory application of FOCAL's energy function to a more realistic 3D setting: optimizing camera parameters in a virtual environment (modeled by a Gaussian splat (Kerbl et al., 2023)). This setting models a robotic agent navigating the scene. It has 6 degrees-of-freedom: 3 translational axes (x,y,z), and 3 rotations (yaw, pitch, roll). The camera then moves around the virtual scene, searching for the view that optimizes FOCAL energy.

We pick 8 scenes (Figure 10) and use the same translation ([−3, 3]) and rotation range (full rotations) for each scene. We use ImageNet classes to compute CLIP energy and minimize it using 150 iterations of BO. We also found that using 450 initial random points can help exploration but is unnecessary for most scenes. We hope to study the optimization schedules more systematically in the future.

We find that this process leads the camera to focus on salient objects in typical angles (like in our previous experiments) in Figure 10. We note that this setting is a significantly more complex "transformation" than classic examples like 2D rotation. Each camera pose sees objects from different

viewpoints, distances, and with varying reflections. While this experiment is preliminary and the process is not yet fully reliable, it shows FOCAL's generalizability.

## 4. Related Work

**Training-time data augmentation**: Data augmentation during training is the simplest and most popular way to achieve invariance. Recent work, such as VIAT (Ruan et al., 2023), ViewFool (Dong et al., 2022), and Ruan et al. (2024), use adversarial viewpoints as augmentations during training/fine-tuning. However, it requires fixing the transformations ahead-of-time, and thus adapting an existing model (e.g., CLIP (Radford et al., 2021)) to new transformations incurs expensive re-training or fine-tuning costs. Additionally, the range of augmentations (e.g., rotation degrees) is unknown and artificially chosen, which can hurt accuracy for some classes (Bouchacourt et al., 2021; Kirichenko et al., 2024). Furthermore, the resulting model is not as robust for classes with fewer training examples (Zhou et al., 2022), making this approach unfit for imbalanced datasets (i.e., most modern datasets like LAION-5B (Schuhmann et al., 2022)). In contrast, test-time approaches like ours provide reliable invariance regardless of the training dataset.

**Equivariant networks**: Architectures like CNNs (LeCun et al., 1999; Fukushima, 1988) and group-equivariant networks (Cohen & Welling, 2016; Cohen et al., 2019; Kondor & Trivedi, 2018; Bronstein et al., 2021) hardcode symmetry priors. While effective for fixed groups (e.g., 2D rotations) or 3D point clouds (Deng et al., 2021), they fail

for complex transformations lacking group structure (e.g., viewpoint changes). Our approach imposes no architectural constraints, enabling broader applicability.

**Learned invariance**: Recent methods aim to learn the invariances from the data by learning augmentation ranges per transformation (Augerino (Benton et al., 2020)), per instance (InstaAug (Miao et al., 2022)), or via normalizing flows (Singhal et al., 2023; Allingham et al., 2024). However, they remain tied to training data and cannot adapt post-hoc. In contrast, our method extracts invariance from foundation models without dataset-specific training.

**Learned canonicalization**: Learned canonicalization has roots in mental rotation (Shepard & Metzler, 1971; Hock & Tromley, 1978). Tarr & Pinker (1989) drew ties between mental rotation and invariant object recognition. These works suggest that canonicalizing can robustly align neural networks to the adaptable nature of the human brain.

These developments have inspired progress in deep learning architectures (Jaderberg et al., 2015; Boominathan et al., 2016). More recently, Kaba et al. (2022) propose a learned canonicalization (LC) approach via minimizing a learned energy function. At test time, it minimizes this function to canonicalize the inputs. PRLC (Mondal et al., 2023) adds a regularization prior to align the canonical with the train set distribution for better accuracy.

However, LC and PRLC still require training specific to the dataset and transform, and do not generalize well beyond their trained settings (Section 3). In contrast, our approach makes no such assumptions; instead, it leverages pre-trained foundation models. Furthermore, we show promising results on significantly more complex transformations.

**3D robustness and pose estimation**: Existing approaches for 3D robustness pool features across multiple views (Fan et al., 2024; Su et al., 2015; Yang & Wang, 2019; Wei et al., 2020; 2022; Hamdi et al., 2021; Kanezaki et al., 2018), whereas our technique only requires one view at test time. Chen et al. (2020) learns category-level pose estimation using analysis-by-synthesis. This approach is related to our approach; however, it is category-specific, whereas our model is category-agnostic. ImageNet3D (Ma et al., 2024) annotates a large dataset of 3D objects with poses for open-set pose estimation, whereas our method is unsupervised.

**Out-of-distribution detection**: Energy-based models have previously been used for OOD detection (Hendrycks & Gimpel, 2017; Liu et al., 2020; Lee et al., 2018; Liang et al., 2018; Graham et al., 2023), but this capability has yet to be harnessed for invariance. To our knowledge, this work is the first to leverage large-scale generative models in conjunction with Equation (1) to create provably invariant models.

## 5. Limitations and Future Work

FOCAL demonstrates strong performance and broad applicability across a variety of transformations, including viewpoint, lighting, and environmental changes. FOCAL draws inspiration from mental rotation and test-time scaling, extracting priors contained in internet-scale foundation models to perform test-time alignment to visually-typical version. FOCAL is thus a significant step forward towards robust perception, with two main limitations and directions for future work that we discuss here: speeding up FOCAL and automatically selecting the transformation.

**Speeding up FOCAL.** Evaluating the energy function for many candidates is computationally expensive (Appendix A.10). Specifically, the complexity is $N \times (C_{\text{transform}} + C_{\text{energy}} + C_{\text{inference}})$, where $N$ is the number of transforms. This limitation is similar to recent LLMs that use test-time search for better reasoning and robustness. Future work could explore a System-1/2 scheme where canonicalization is only used when necessary, enabling users to gain the robustness benefits of FOCAL without incurring as much of a cost in runtime efficiency. As a preliminary experiment, in 2D, we were able to detect upright vs. non-upright images with 95% accuracy by comparing the input against $+90°$ and $-90°$ and thresholding the energy difference.

**Selecting the Transformation.** While FOCAL shows great progress in achieving robust visual perception across a wide variety of transformations at test-time, it remains an open question on how to select which transformation generator(s) to use. Future work could explore how to automatically determine which generator(s) to use at test-time, making FOCAL even more easily applicable to real-world settings.

## 6. Conclusion

FOCAL offers a test-time, scalable, data-driven strategy for robust visual perception. FOCAL takes inspiration from canonicalization, projecting out-of-distribution inputs to the most visually-typical view. By leveraging the priors learned by foundation models trained on internet-scale image datasets, FOCAL's "vary and rank" scheme enables us to optimize for visually-typical views that are suitable for downstream models. By challenging the prevailing assumption that invariance requires dataset-specific training or architectural compromises, FOCAL offers a new, test-time path toward robust perception for embodied agents facing a wide variety of transformations.

**Acknowledgements.** This project was supported, in part, by Beyster Fellowship to R. Feng, by NSF 2215542, NSF 2313151, and Bosch gift funds to S. Yu at UC Berkeley and the University of Michigan.

## Impact Statement

This paper presents work whose goal is to advance the field of machine learning. Specifically, this paper aims to have a positive impact on the robustness of machine learning models. Our test-time approach may make models more reliable and trustworthy on out-of-distribution transformations.

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

## Appendix Summary

This appendix provides extended results, analyses, and implementation details supporting the main paper. The content is organized as follows:

- **Additional Results and Figures**

    - *3D CO3D examples*: Shows examples of canonicalized frames in CO3D (Figure 11)
    - *3D Results*: Additional results showing while TRELLIS can canonicalize to an extent on inputs near its training distribution, FoCAL still outperforms it (Figure 12, Table 1, Table 2)
    - *2D Classification and Pose Estimation*: Demonstrates improved accuracy and pose estimation across CIFAR10/100/STL10 (Figure 13, Table 3)
    - *Segmentation Results*: Matches PRLC's COCO mAP while achieving +2.1% pose accuracy (Table 4)
    - *ImageNet Generalization*: Outperforms PRLC by up to +11.4% on rotated inputs (Table 5)
    - *Cross-Dataset Analysis*: Shows $< 3\%$ variance in pose error vs PRLC's 12–18% drops (Figure 14)
    - *Ablations*: Validates energy function design and compares against TTA (Table 6, Table 7).
    - *DINOv2 Contrast results*: Evaluating FoCAL with DINOv2 shows similar trends to CLIP (Figure 16)
    - *Color Correction Results*: Shows direct comparison against stronger supervised color correction baselines (Appendix A.9)
    - *Computational Complexity Analysis*: Discussion on FLOPs and Runtime (Table 8)

- **Experimental Setup**

    - 3D protocols with CO3D/Objaverse-LVIS
    - Color transformation formalization
    - Bayesian hyperparameter optimization details

## A. Additional Results and Figures

### A.1. 3D CO3D Examples

In Figure 11, we show some examples of CO3D (Reizenstein et al., 2021) frames being corrected through FoCAL, comparing the original input views that were incorrectly classified by OV-Seg (Liang et al., 2023) and their canonicalized views that are correctly classified by OV-Seg.

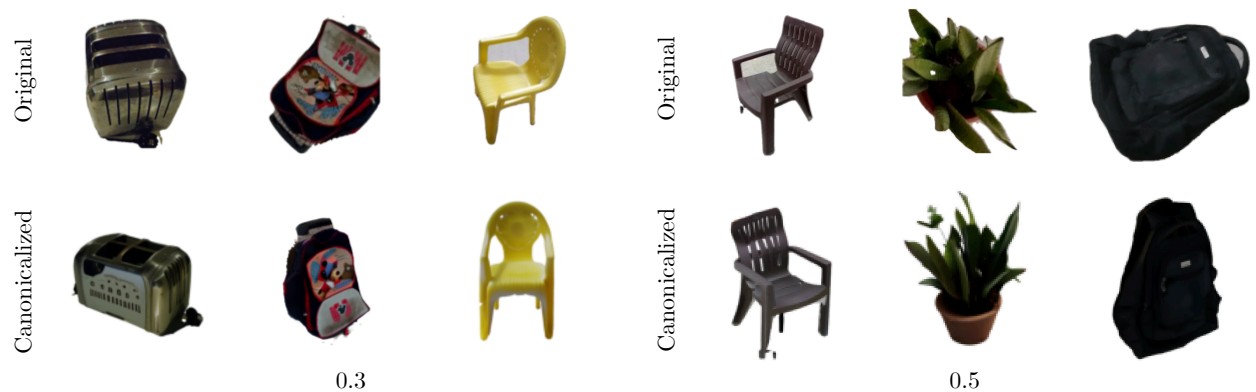

*Figure 11.* **Viewpoint canonicalization improves accuracy by improving viewpoint robustness on CO3D:** Example renderings from filtered CO3D before (original) and after (canonicalized) applying FoCAL with an original ground-truth probability threshold of 0.3 and 0.5. The canonicalized views provide more informative perspectives for object recognition.

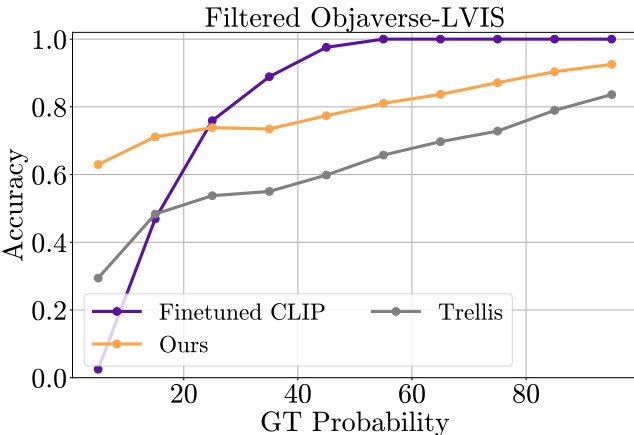

*Figure 12.* Accuracy over GT probability. For each object in Objaverse, we take a random render and evaluate FoCal vs. OV-Seg and a TRELLIS baseline along bins of GT label probability. We evaluate FoCal with 5 pretilts at intervals of 45 degrees. We find that FoCal still consistently outperforms the TRELLIS baseline.

| Threshold | OV-Seg (Finetuned CLIP) | Random Rotation | TRELLIS | Ours |
|:---:|:---:|:---:|:---:|:---:|
| 0.3 | 5.71% | 38.8% | 45.9% | **49.5%** **(+3.6%)** |
| 0.5 | 28.5% | 46.6% | 53.4% | **55.3%** **(+1.9%)** |

*Table 1.* Comparison of FoCal against TRELLIS as a canonicalizer and random rotations. Accuracy is reported for each setting on quality-filtered CO3D frames, where for each video, we randomly sample a frame with a ground truth label probability below two different difficulty thresholds (0.3, 0.5) and filter it as described in Appendix B.1 for image and segmentation quality. While surprisingly, random rotation and TRELLIS perform reasonably well, FoCal outperforms both. One possible explanation is that CO3D primarily contains horizontal orbits from above the object, limiting tilt variation in its viewpoints.

### A.2. 3D Results

We also found that TRELLIS (Xiang et al., 2024) itself can be used as a canonicalizer in limited settings, where the input images are near its training distribution. TRELLIS was trained on Objaverse (Deitke et al., 2023) views with varying elevation and azimuth (no tilt) and implicitly uprights objects when creating the 3D assets it generates. We can then select a particular view and stick with it everytime, creating a new point of comparison with TRELLIS. We found that 30 degrees elevation and 150 degrees azimuth works the best.

To better understand FoCal's performance vs. TRELLIS (Xiang et al., 2024), we take each filtered Objaverse-LVIS (Deitke et al., 2023) object we evaluated in Section 3 and select a random view. We then randomly tilt the image by a random 45 degree angle to ensure some distributional shift from TRELLIS' training data. To adapt FoCal to handle the additional tilt axis, we perform 2D canonicalization as in Section 3 and then attempt 5 angles at 45 degree intervals centered at the selected 2D canonical angle, and take the minimum energy viewpoint over all the examined images. Finding that a majority of good images tend to be at elevation 30, we explore 12 azimuths at an interval of 30 degrees that this elevation.

We plot the accuracy of the three methods (plain OV-Seg (Liang et al., 2023), TRELLIS, and FoCal) binned by the GT probability of the input view in Figure 12. We find that our approach consistently outperforms TRELLIS at all GT probability bins. Like Figure 5, FoCal outperforms OV-Seg on bad viewpoints with some decrease on good viewpoints.

Next, we further analyze our results on CO3D (Reizenstein et al., 2021), including using TRELLIS (Xiang et al., 2024). We also compare the performance against selecting a randomly rotated (in 2D) image. The results on the filtered and thresholded data splits from Section 3 are shown in Table 1.

We find that FoCal outperforms random rotations and TRELLIS as a canonicalizer, but random rotations and TRELLIS still provide surprisingly strong performance. FoCal provides a 3.6% gain over TRELLIS on the 0.3 threshold and a 1.9% gain over TRELLIS on the 0.5 threshold. One possible explanation for this is that CO3D primarily contains horizontal orbits from above the object, limiting variation in its viewpoints. Because of this, there is also likely less tilt variation, keeping the

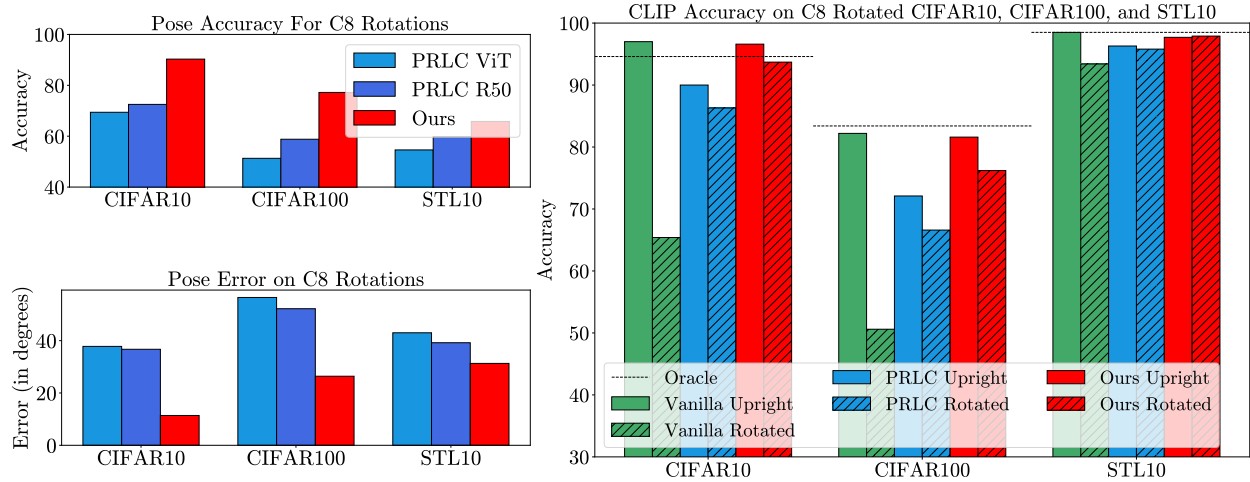

*Figure 13.* **FOCAL beats PRLC dataset specialist canonicalizers on rotated accuracy and pose estimation.** **(a)** We show that FOCAL achieves better pose accuracy than PRLC on their datasets. **(b)** We demonstrate that FOCAL outperforms PRLC dataset specialist models in terms of pose error. **(c)** As a result of our superior pose estimation, we generalize better to new models like CLIP. Dashed lines represent oracle performance, i.e., perfectly undoing the rotation except for any loss of corner information from cropping. This result highlights FOCAL's strong canonicalization ability.

poses to in-distribution poses. As for random rotation, depending on the sharpness of the angle above 2D rotations may effectively mimic moving around the orbit. Thresholding objects for hardness may also lead towards "adversarial" examples that lie in an extreme local extrema. We leave further investigation to future work.

Finally, we compare test-time augmentation (TTA) against FOCAL on 3D viewpoints. We compare against a TTA strategy that averages over 1, 5, and 10 random transformations out of the 60 generated TRELLIS (Xiang et al., 2024) renders. To match FOCAL, we also blur the logits before applying TTA (Appendix B.1). We also compare FOCAL with a variant of FOCAL that averages the logits over the top 5 (and top 10) ranked viewpoints. We evaluate on Objaverse-LVIS (Deitke et al., 2023) and report the results in Table 2. We find that FOCAL with averaging over the top 10 ranks performs the best on average at 84.5%, a best view performance of 93.4%, and a worst view performance of 71.3%, where best and worst view are defined as the first and last ranks of the quality-filtered Objaverse-LVIS dataset as in Figure 5. FOCAL performs especially well on the best view. TTA also performs well, achieving some good performance on the worst view (although less than ours at equivalent numbers of views), perhaps owing to the nature of 3D consisting of more viable in-distribution viewpoints than a setting like 2D rotation.

| Method | # Views | Mean | Best View | Worst View |
|--------|---------|------|-----------|------------|
| Ours | 1 | 79.5% | 93.3% | 61.6% |
| Ours | 5 | 84.1% | 93.9% | 70.4 |
| Ours | 10 | 84.5% | 93.4% | 71.3% |
| TTA | 1 | 67.8% | 75.3% | 56.2% |
| TTA | 5 | 75.2% | 83.9% | 62.6% |
| TTA | 10 | 76.4% | 85.1% | 63.7% |

*Table 2.* Comparison of FOCAL vs. TTA when averaging over multiple views. For the same number of views, our method achieves strictly better accuracy in each category (best, worst, mean).

### A.3. 2D Classification and Pose Estimation

FOCAL demonstrates superior canonicalization capabilities across three key 2D benchmarks. On CIFAR10/100 (Krizhevsky et al., 2010) and STL10 (Coates et al., 2011), our method achieves consistent improvements over PRLC's (Mondal et al., 2023) dataset-specific canonicalizers: +0.3-2.3% upright accuracy gains (Table 3), +0.4-2.1% robustness to random C8

| Pretrained Network | | ResNet50 (PRLC-Trained) | | ViT (PRLC-Trained) | |
|---|---|---|---|---|---|
| Datasets | Canoncalizer | Accuracy | Random Rotation (*C8*) | Accuracy | Random Rotation (*C8*) |
| **CIFAR10** | Rotation Aug. | 94.9 | 90.1 | 96.3 | 93.7 |
| | PRLC | 96.1 | 95.1 | 95.8 | 94.8 |
| | Ours | **96.5 (+0.4%)** | **95.6 (+0.5%)** | **97.4 (+1.1%)** | **96.0 (+1.2%)** |
| | *Oracle* | *96.6* | *95.9* | *97.6* | *96.7* |
| **CIFAR100** | Rotation Aug. | 80.2 | 74.1 | 82.6 | 78.4 |
| | PRLC | 83.1 | 81.8 | 83.9 | 82.2 |
| | Ours | **83.8 (+0.7%)** | **82.2 (+0.4%)** | **86.3 (+2.4%)** | **84.4 (+2.2%)** |
| | *Oracle* | *84.4* | *83.4* | *87.1* | *85.5* |
| **STL10** | Rotation Aug. | **98.1** | 95.0 | **97.9** | 94.1 |
| | PRLC | 95.2 | 94.1 | 95.7 | 93.9 |
| | Ours | 96.2 (-1.9%) | **95.6 (+0.6%)** | 96.0 (-1.9%) | **95.2 (+1.1%)** |
| | *Oracle* | *97.4* | *96.7* | *97.3* | *96.4* |

*Table 3.* **FOCAL beats PRLC dataset specialist canonicalizers and models on rotated accuracy.** We find that FOCAL outperforms PRLC, without any training, across all PRLC specific model and dataset pairs on both upright inputs and randomly rotated inputs. We compare against just upright images in the Accuracy columns. Oracle refers to a system where the exact angle to upright is known, and thus only measures the change in accuracy due to loss of information due to rotating, cropping, and re-rotating. Random Rotation (*C8*) applies a random *C*8 transform to the input before passing it to the aligner / model. Best non-oracle rows are bolded. Rotation Augmentation numbers taken from (Mondal et al., 2023), and the rest are reproduced using their provided code and default hyperparameters. This result highlights that we can beat PRLC even on their best settings.

| Pretrained Network | | SAM | |
|---|---|---|---|
| Dataset | Canonizalizer | mAP Random Rotation (C4) (%) | Pose Accuracy (%) |
| COCO | Naive | 62.1 | - |
| | PRLC | 65.9 | 86.8 |
| | Ours | 65.9 | **88.9 (+2.1%)** |
| | *Oracle* | *66.3* | *-* |

*Table 4.* **FOCAL matches PRLC on segmentation**: We first report FOCAL's performance on mAP on COCO, PRLC without any training (left). We then show that the *C*4 pose accuracy is higher than PRLC by 2.1%. This shows FOCAL's ability to generalize to segmentation without supervision. All numbers are reproduced using Mondal et al. (2023)'s pre-trained checkpoint.

rotations (Table 3), lower pose error and higher pose accuracy (Figure 13).

Notably, FOCAL approaches oracle performance gaps within 0.9-2.1% across all metrics, suggesting our energy minimization framework effectively approximates ideal canonicalization despite unknown rotation angles. Our method achieves this without any supervision or dataset/task-specific training.

### A.4. Segmentation Results

FOCAL's geometric canonicalization transfers seamlessly to segmentation tasks without segmentation-specific training. On COCO (Lin et al., 2014), FOCAL: (1) Matches PRLC's 65.9 mAP despite PRLC being trained specifically on segmentation data whereas FOCAL is zero-shot; (2) Achieves 88.9% pose accuracy (+2.1% over PRLC); (3) Nears oracle mAP (66.3 vs 66.9). This demonstrates that our approach learns fundamental viewpoint normalization rather than task-specific artifacts. The pose accuracy gains directly translate to better segmentation consistency across rotations, as evidenced by the mAP parity despite PRLC's segmentation-aware training (Table 4).

| ImageNet | ResNet50 (Vanilla-Trained) | | ViT (Vanilla-Trained) | | CLIP (Vanilla-Trained) | |
|---|---|---|---|---|---|---|
| Canonicalizer | Accuracy | Random Rot. (C8) | Accuracy | Random Rot. (C8) | Accuracy | Random Rot. (C8) |
| None | 73.3 | 49.0 | 79.4 | 58.8 | 77.1 | 67.0 |
| PRLC* | 63.1 | 59.2 | 63.7 | 60.5 | 72.1 | 69.6 |
| Ours | **64.4 (+1.1)** | **62.4 (+3.2)** | **72.7 (+9.0)** | **71.2 (+11.4)** | **75.2 (+3.1)** | **73.9 (+4.3)** |
| *Oracle* | *73.3* | *70.8* | *79.4* | *77.4* | *77.1* | *75.3* |

*Table 5.* **FOCAL generalizes better to ImageNet and outperforms PRLC's canonicalizers.** We find that FOCAL outperforms PRLC on ImageNet, without any training, on both upright inputs and randomly rotated inputs. We compare against just upright images in the Accuracy columns. Oracle refers to a system where the exact angle to upright is known, and thus only measures the change in accuracy due to loss of information due to rotating, cropping, and re-rotating. Random Rot. ($C8$) applies a random $C8$ transform to the input before passing it to the aligner / model. Best non-oracle rows on rotated performance are bolded. For PRLC, the canonicalizers were the best performing ones from other datasets (STL10 for both ResNet50 and ViT). Thus, they were not trained specifically for ImageNet.

### A.5. ImageNet Generalization

FOCAL shows remarkable cross-dataset generalization on ImageNet (Deng et al., 2009): +11.4% rotated accuracy for ViT (Dosovitskiy et al., 2021) vs PRLC (Mondal et al., 2023) (71.9 vs 60.5%); +4.3% absolute gain for ResNet50 (He et al., 2016) under rotation; CLIP (Radford et al., 2021) performance improves +4.4% under rotation.

Notably, PRLC canonicalizers trained on small datasets (STL10/CIFAR) degrade significantly on ImageNet (-16.7% ResNet50 upright accuracy vs ours), failing to generalize outside their training setting. FOCAL, however, maintains strong performance through its dataset-agnostic energy formulation.

### A.6. Cross-Dataset and Model Analysis

FOCAL 's unified framework enables superior cross-domain transfer compared to PRLC's (Mondal et al., 2023) specialized canonicalizers. When transferring across datasets, PRLC suffers 12-18% drops in pose accuracy (Figure 14c), while our method maintains < 3% variance in pose error. Transferring to CLIP (Radford et al., 2021) yields particularly strong results, with +15% relative accuracy gains over PRLC (Figure 15). These results underscore a key advantage: by avoiding dataset-specific training, FOCAL develops general canonicalization strategies that transfer seamlessly across both architectures (ResNet50/ViT/CLIP) and data distributions (CIFAR/STL10/ImageNet).

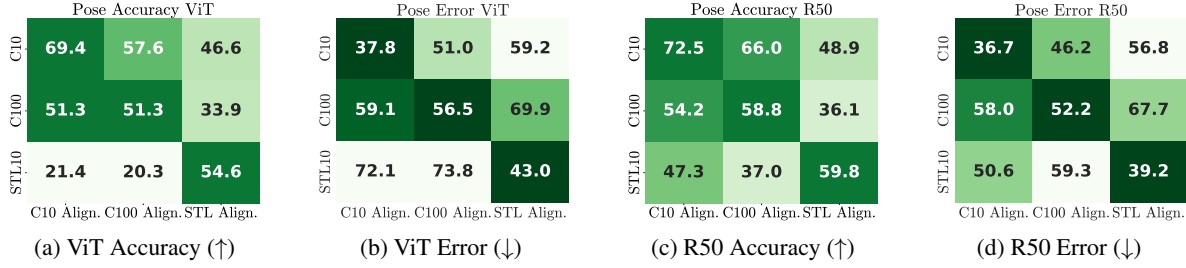

*Figure 14.* **FOCAL generalizes better across datasets when mixing up aligners and downstream models**. PRLC performance on pose estimation drops significantly when using a canonicalizer trained from a different dataset compared to FOCAL, which applies one technique across all settings. This result highlights the generalizability across datasets of an unsupervised approach.

### A.7. Ablation Studies

Component ablations validate critical design choices in our energy formulation. Combining CLIP and diffusion energies reduces pose error by 64% compared to CLIP alone (13.5° vs 37.1°) (Table 6), while test-time augmentation (TTA) underperforms by 10-14% on CIFAR benchmarks (Table 7). These experiments confirm that naive augmentation cannot substitute learned canonicalization and that multi-energy fusion provides complementary benefits.

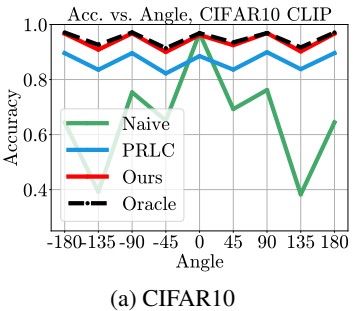
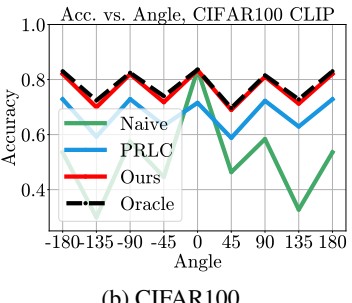
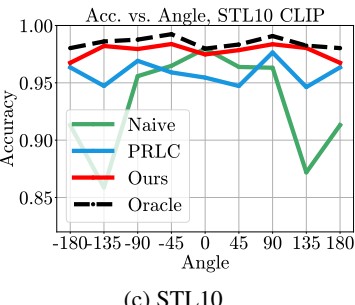

(a) CIFAR10       (b) CIFAR100       (c) STL10

*Figure 15.* Accuracy vs. $C8$ angle on CLIP. Like on ResNet50, we find that using FOCAL leads to invariant predictions over angles, outperforming PRLC. The contrast is particularly clear for CLIP on CIFAR10 and CIFAR100, where our accuracy over angle is consistently above PRLC.

| Method | Pose Accuracy | Avg. Pose Error (degrees) |
|---|---|---|
| Only CLIP | 68.9% | 37.1° |
| Only diff | 82.7% | 22.6° |
| diff+clip | 89.5% | 13.5° |

*Table 6.* Ablation study on energy function components for pose estimation. The combination of diffusion (diff) and CLIP achieves the highest accuracy and lowest pose error.

| | CIFAR10 | CIFAR100 | STL10 |
|---|---|---|---|
| No uprighting | 65.4 | 50.6 | 93.4 |
| Ours | 93.7 | 76.2 | 97.5 |
| TTA | 82.8 (-10.9) | 61.7 (-14.5) | 96.6 (-0.9) |

*Table 7.* Comparison of CLIP's accuracy across different datasets using No Uprighting, FOCAL, and Test-Time Augmentation (TTA). Our method significantly outperforms TTA, especially for larger invariance ranges like C8.

## A.8. Contrast Canonicalization for DINOv2

We also provide contrast results for FOCAL on DINOv2 in Figure 16 (similar to CLIP in the main paper Figure 7). It follows the same trend, achieving significant accuracy gains on the largest transformations.

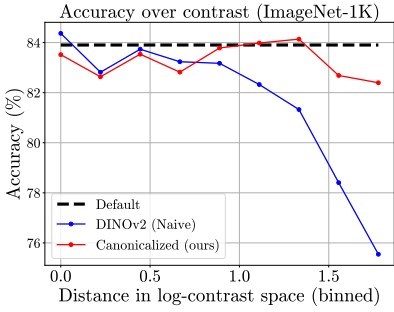

*Figure 16.* Contrast results for DINOv2

## A.9. Comparison against Color Correction baselines

We benchmark our method on the RCC dataset and compare it against Barron & Tsai (2017) and the gray world method. We search for the two log-chrominance values in the range $[-0.7, -0.3]$, and used both classifier and diffusion energies with the default values for classifier energy, and a coefficient of $-10$ for diffusion energy, with the diffusion energy evaluated on

a single step $t = 50$. We used BO to optimize the color using 10 random initial points and 50 BO iterations. Our method only produces visually typical (in-distribution) images, not necessarily the most color-neutral version, and thus, performs poorly on this test compared to baselines designed to achieve high color accuracy on this task (Figure 17):. Specifically, we achieve a median angular error of $6.4°$ compared to Barron & Tsai (2017)'s $1.3°$, though we do beat gray-world's $9.97°$. Our method is nowhere near the specialized state-of-the-art methods on this task, but ours is a more general algorithm designed to improve invariance. We leave it to future work to improve the base performance of FOCAL against supervised color approaches.

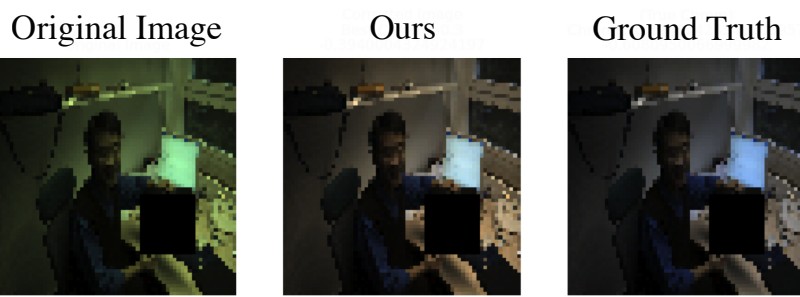

*Figure 17.* Example image of color correction from the RCC Dataset. Our method (center) produces an image with typical lighting, which is not necessarily the same as completely neutral white colors (right).

### A.10. Computational Complexity Analysis

While our method primarily focuses on improving generalization through equivariant learning, we provide a detailed analysis of its computational costs. The overall complexity can be decomposed as:

$$\text{Naive Cost} = C_{\text{inference}}$$
$$\text{TTA Cost} = N \times C_{\text{inference}}$$
$$\text{Our Cost} = N \times (C_{\text{transform}} + C_{\text{energy}} + C_{\text{inference}})$$

where $N$ represents the number of evaluated transformations, $C_{\text{transform}}$ denotes transformation cost, $C_{\text{evaluation}}$ encompasses CLIP and diffusion forward passes, and $C_{\text{inference}}$ is the final prediction cost. In principle, inference only has to be done once (after uprighting), so the $C_{\text{inference}}$ does not need to be multiplied by $N$. We use the current cost as an upper bound to compare the cost more easily against TTA. We also assume that system-1/2 methods to skip unnecessary canonicalization are not being used. This setting represents the worst-case scenario for our method from computational cost perspective.

**Example calculation** For 2D rotation with $N = 8$ orientations and $K = 5$ diffusion steps, and CLIP as classifier:

$$\text{Standard inference}: \quad 1 \times (\text{CLIP})$$
$$\text{TTA}: \quad 8 \times (\text{Rotation} + \text{CLIP})$$
$$\text{Our method}: \quad 8 \times (\text{Rotation} + \text{CLIP} + 5 \times \text{Diffusion})$$

Assuming image rotation uses negligible FLOPs, we get: $1\times$ for naive, $\approx 8\times$ for TTA, $\approx 56\times$ for ours.

**Latency reduction through parallel inference** Despite increased FLOPs, our method is parallelizable since the energy functions can be evaluated independently on different points, and even different diffusion steps can be evaluated in parallel (for energy computation only). Thus, the theoretical latency under parallelism is only:

$$\text{Theoretical Latency (parallel)} = \max (\text{Classifier}, \text{Diffusion step}, \text{CLIP})$$

**FLOPs and runtimes for our experiments** Here, we detail the FLOPs and runtime costs for key experimental settings used in this paper. We use the DeepSpeed profiler and note the FLOPs, total cost, and runtime/latency (unoptimized). All

experiments were done on an RTX 2080Ti GPU except 3D viewpoint, which was done on an RTX 6000 Ada Generation GPU. FLOPs for 3D are omitted due to compatibility issues with DeepSpeed and TRELLIS. 45% of the average 3D runtime was spent generating the 3D asset in TRELLIS.

*Table 8.* Per-Experiment approximate FLOP Costs and Runtime latency (unoptimized)

| Experiment | #Transforms (N) | Diffusion Steps | Baseline FLOPs (T) | TTA FLOPs (T) | FOCAL FLOPs (T) | Runtime (s/it) |
|---|---|---|---|---|---|---|
| 2D Rotation (C10/C100/IN) | 8 | 0 | 0.33 | 2.6 | 2.6 | 0.47 |
| 2D Rotation (COCO) | 4 | 10 | 0.33 | 1.29 | 19.1 | 0.94 |
| Color (C100/IN) | 35 | 0 | 0.33 | 11.6 | 11.6 | 5.2 |
| Color (RCC) | 60 | 1 | 0.33 | 19.8 | 46.2 | 10.3 |
| Contrast (C100/IN) | 12 | 0 | 0.33 | 3.97 | 3.97 | 1.22 |
| Day-Night | 2 | 0 | - | - | 0.66 | 0.16 |
| Active vision | 500 | 0 | - | - | 49.5 | 238.1 |
| 3D Viewpoint | 61 | 5 | - | - | - | 13.3 |

# B. Experimental Setup

## B.1. Experimental Setup - 3D

**Dataset Details:** The Objaverse-LVIS (Deitke et al., 2023) dataset contains 46207 3D assets labeled with one of 1156 categories from LVIS (Gupta et al., 2019). From these 3D assets, multiview images can then be generated with software such as Blender (Blender Foundation, 2022). The CO3D dataset contains a collection of 18619 real multiview video sequences of 51 MS-COCO (Lin et al., 2014) objects with each video recording the object in a circular orbit, along with segmentation masks for each frame.

**Rendering:** For Objaverse-LVIS (Deitke et al., 2023), we generate our base input renders at viewpoints in the upper viewing hemisphere. We sample at an interval of 30 degrees and a radius of 2.2. We offset the views by 10 degrees to avoid perfectly aligned views that eliminate critical 3D context of the object. This leads to 36 renders in total.

For Objaverse-LVIS (Deitke et al., 2023) in the "vary" stage we sample 60 viewpoints. These are at every 30 degrees of elevation and azimuth, for a grid of $[-60, -30, 0, 30, 60] \times [-180, -150, -120, -90, -60, -30, 0, 30, 60, 90, 120, 150, 180]$. For CO3D (Reizenstein et al., 2021), we sample the 12 viewpoints at the elevation 30 and azimuths of $[-180, -150, -120, -90, -60, -30, 0, 30, 60, 90, 120, 150, 180]$.

**Filtering:** Due to Objaverse-LVIS' (Deitke et al., 2023) label quality concerns primarily stemming from the existence of multiple similar labels (e.g., orange vs. mandarin orange vs. tangerine, ring vs. wedding ring, etc.) we filter out any objects that: 1) had fewer than 10% of its renders classified correctly or 2) lacked a clear winner class that was predicted at least 33% more than the 2nd most common class. This results in a test set of 14789 objects.

For Objaverse-LVIS in Fig. 5, we evaluate FOCAL on the 0th, 12th, 24th, and 35th ranked viewpoints, which corresponds to the 0th, 33rd, 66th, and 100th percentile.

For CO3D (Reizenstein et al., 2021), we focus on videos which contain sufficient viewpoint variation to induce errors in classification. Specifically, we filter for videos that have at least one frame with a probability greater than 80% for the ground truth class and at least one frame with a probability less than a threshold $t$. We run with two thresholds: $t = 0.3$ and $t = 0.5$. We then pick a random frame of probability less than $t$ for each video. For $t = 0.3$, this gives us 11157 frames, and for $t = 0.5$, this gives us 12186 frames.

With these frames, we then filter out the frames for segmentation and image quality. To do this, we crop the objects based on their segmentation mask's bounding box, multiply the size of the crop by 1.2, and resize the crop to 518 x 518 (following the preprocessing of TRELLIS (Xiang et al., 2024). We then pass the crop and the cropped segmentation to gpt-4o-mini-2025-04-16 (OpenAI, 2025) with the following prompt:

> **Prompt**
>
> You are evaluating an image (left) and its segmentation overlay (right)
> Criteria:
> 1) There is a single, clearly visible object that fits completely in the frame.
> 2) The main object should not have any parts outside the frame. There should be a clear margin on all sides.
> 3) The main object should be easily distinguishable and not blurry.
> 4) The segmentation mask should accurately outline the main object.
>
> If all criteria are met, respond 'PASS' plus the main object. Otherwise, respond 'FAIL' plus a short reason.
> Your answer must start with 'PASS' or 'FAIL' and use fewer than 10 words.

This filtering results in 1504 frames (13.5%) for $t = 0.3$, and 1865 frames (15.3%) for $t = 0.5$.

**Calculating Energy:** For both Objaverse (Deitke et al., 2023) and CO3D (Reizenstein et al., 2021), we use $\alpha = 1, \beta = 0.5$ following the 2D experiments (B.3). We also used the diffusion energy (steps 500 to 1000 with stride 100) with a factor of 5. We also normalized the CLIP (Radford et al., 2021) energy with a normalizing prompt of "a photo of an object on a bright white backdrop." and a temperature of 0.5 to adjust CLIP to the background removed images used in these experiments.

### B.2. Experimental Setup – Color and Contrast

We define the color shift transformation using the popular von Kries model (KRIES, 1905) where an illuminant vector with the RGB values $L = [L_R, L_G, L_B] \in \mathbb{R}^3$ is multiplied element-wise with every pixel in the image. We then generate this illuminant vector $L$ by sampling in the log-chrominance space (Barron & Tsai, 2017). Specifically,

$$L_u, L_v \sim U[-1, 1] \tag{6}$$

$$[L_R, L_G, L_B] = [\frac{\exp(-L_u)}{z}, \frac{1}{z}, \frac{\exp(-L_v)}{z}] \tag{7}$$

where $z = \sqrt{\exp(-L_u)^2 + \exp(-L_v^2) + 1}$ is a normalizing constant and $L_u, L_v$ are the log-chroma values sampled from the uniform distribution with range $[-1, 1]$. Intuitively, the log-chroma space defines the $R/G$ and $B/G$ ratios in log-space. A range of $[-1, 1]$ corresponds roughly to a $7\times$ change in the ratio between the minimum and maximum points of the range.

We define the contrast as a gamma transformation $x^\gamma$, where the log of the gamma is sampled randomly at uniform $\log(\gamma) \sim U[-2, 2]$. This means gamma lies between $e^{-2}$ and $e^2$.

We optimize the energy functions using Bayesian optimization. For initialization, we use random as well as a grid of initial samples. Color uses a uniform grid of $3 \times 3$, 6 random points, and 20 iterations. Contrast uses 3 grid points, 4 random points, and 5 iterations.

### B.3. Hyperparameters for the energy functions

For experiments on ImageNet, CIFAR10, CIFAR100, and STL10, we only used the classification energy for computational efficiency. We used $\alpha = 1, \beta = 0.5$ for all these settings. In practice, a wide range of $\beta$ works well. The same is true for active vision experiments, but the figures shown in the paper used $\beta = 0.33$.

For segmentation, we used the diffusion energy (steps 50 to 150 with stride 10) with a factor of 0.67 along with CLIP energy factor of 0.54 and $\beta = 0.2$.

All these hyperparameters were found using Bayesian Optimization with the same kernel and acquisition function mentioned in Section 2.4 and performed using the Bayesian Optimization Toolbox (Nogueira, 2014) for 300 time steps. Each energy hyperparameter was tuned on a small training or validation set by recording logits and finding the combination of energy functions that maximized accuracy.

