# OpenReview forum: "Test-Time Canonicalization by Foundation Models for Robust Perception"
_ICML.cc/2025/Conference — ICML 2025 poster_

### Official Review · Reviewer_snMu · 2025-03-12

**Overall Recommendation:** 4

**Summary:**

The paper introduces FoCAL, a zero-shot framework for achieving approximate invariance to complex transformations at scale without requiring additional training. FoCAL operates in two steps: (1) generating multiple transformed variations of an input and (2) ranking them using energy functions derived from CLIP and Stable Diffusion to select the most "canonical" version. The method is evaluated on transformations such as 3D viewpoint shifts, lighting changes, and environmental variations across multiple datasets, including ImageNet, COCO, Objaverse-LVIS, and CO3D.

**Claims And Evidence:**

The claims made in the submission are generally supported by convincing experimental results. However, as mentioned in the *Methods And Evaluation Criteria* section, additional results could better demonstrate the advantages or disadvantages of their method in comparison.

**Essential References Not Discussed:**

I don't know any additional essential references.

**Experimental Designs Or Analyses:**

As reported in the *Methods And Evaluation Criteria* section, additional comparison would improve the experimental design of the paper.

**Methods And Evaluation Criteria:**

There are some limitations in the evaluation. For example, the paper does not include comparisons with domain adaptation (DA) methods or equivariant architectures for transformations like rotation, which would have provided a more comprehensive understanding of the method's performance. For example, in Section 4.2: Lighting (color and contrast), the only comparison made is with vanilla CLIP.  Additionally, while the authors mention that their method is not superior to supervised approaches, they do not provide quantitative results or scores to explicitly assess the difference between their method and the supervised approaches.

**Other Comments Or Suggestions:**

Figure 4 is never cited in the paper.

**Other Strengths And Weaknesses:**

The paper is well-written and easy to follow, with clear explanations of the key ideas. The figures are self-explanatory and complement the text, helping to further clarify the proposed approach. Additionally, the authors provide a thoughtful discussion of the method’s limitations and potential future work, which adds transparency to the study. However, one weakness is the lack of comparisons with existing data augmentation methods and equivariant architectures for transformations like rotation. A direct comparison with these methods would help highlight the advantages and limitations of the proposed approach and could strengthen the paper’s effectiveness.

**Questions For Authors:**

* Q1: I understand that FoCAL addresses problems that other methods struggle with (e.g., rare classes for data augmentation and complex real-world transformations for equivariant architectures), but how does it perform in settings where these methods are typically used and perform well?

* Q2: How does the method compare with the one proposed in [1]?

* Q3: In lines 317-318, it is mentioned that FoCAL does not surpass supervised approaches (e.g., Barron & Tsai, 2017; Hernandez-Juarez et al., 2020), but no results are provided. Could the authors provide a comparison with these methods?

-----
[1] Mondal, Arnab Kumar, et al. "Equivariant adaptation of large pretrained models." Advances in Neural Information Processing Systems 36 (2023): 50293-50309.

**Relation To Broader Scientific Literature:**

The key contribution of this paper is to provide a zero-shot framework for achieving approximate invariance to complex transformations such as lighting changes and 3D viewpoint shifts. This approach builds upon existing work, distinguishing itself by using energy functions derived from CLIP and Stable Diffusion to evaluate transformations, without requiring additional training or task-specific fine-tuning. Additionally, the paper relates to the broader work on equivariant architectures, providing evidence that FoCAL can handle complex transformations. FoCAL leverages pre-trained models and applies a ranking mechanism to select the canonical version of transformed inputs, thus offering a more generalizable solution, making the method particularly useful for large-scale, real-world applications where transformation invariance is needed without extensive retraining.

**Theoretical Claims:**

There are no formal proofs in this paper, but the approach is grounded in empirical design choices with inspiration from prior studies.

---

> ### Author Rebuttal · Authors · 2025-04-01
>
> Thank you for your thoughtful review. We are glad you found value in our idea and we hope FOCAL can be a key step towards a zero-shot approach towards a wide range of visual invariance. We address your concerns and questions below:
>
> - **Clarification question**: By DA, are you referring to data augmentation or domain adaptation?
>
>   We use DA for data augmentation in our paper (line 50, right column). Apologies for the confusion. We are not sure what an appropriate comparison against domain adaptation would be. Thus, the rest of this response assumes data augmentation, but if you meant domain adaptation, please clarify what you have in mind.
>
> - **Comparisons on 2D rotation**: Our PRLC [1] experiments (Figure 7 in the main paper, Table 1 in the appendix) indeed show a comparison against [1] as well as against data augmentation.
>
>   - The experimental setting in Table 1 is ideal for data augmentation because it uses balanced datasets (e.g., CIFAR10) with known augmentations (2D rotation). On these datasets (CIFAR10, CIFAR100, STL10), we beat both data augmentation and PRLC [1] baselines.
>
>   - We don’t show explicit comparison against equivariant architectures because we could not find ViT/R50 scale baselines that could be compared fairly, but we would be happy to evaluate any baselines that you suggest.
>
>   - We do have an indirect comparison against equivariant nets through [1] because it uses an equivariant net to canonicalize the images.
>      [1] https://proceedings.neurips.cc/paper_files/paper/2023/file/9d5856318032ef3630cb580f4e24f823-Paper-Conference.pdf
> - **“Figure 4 is never cited in the paper.”**
>
>    - Thanks! Will fix.
> - **“In lines 317-318, it is mentioned that FoCAL does not surpass supervised approaches (e.g., Barron & Tsai, 2017; Hernandez-Juarez et al., 2020), but no results are provided. Could the authors provide a comparison with these methods?”**
>
>   - That’s a fair point. On the RCC (Recommended Color Checker) dataset, Barron & Tsai (2017) achieve a median angle error of 1.3 degrees. The Gray World baseline achieves a median angle error of 9.97 degrees. Ours achieves a median angle error of 7.3 degrees. When it comes to classification, we find that the classification accuracy differences are minute (<0.5%).
>
>   - We will add a more detailed comparison in the paper. We will also try to find supervised baselines for the contrast transformation (or train such baselines ourselves) and add those to the paper as well.

---

> > ### Comment · Reviewer_snMu · 2025-04-02
> >
> > I thank the authors for addressing all my concerns. I also apologize for the confusion regarding the DA. I have adjusted my review accordingly and have raised the score.

---

### Official Review · Reviewer_PWqm · 2025-03-12

**Overall Recommendation:** 4

**Summary:**

The paper proposes to construct an energy measure from outputs of pretrained foundation models. This energy can be minimized over transformations of the input image to obtain a canonicalization of the image. The minimization can be done through bayesian optimization. The experimental results are good.

**Claims And Evidence:**

The authors write
> As a bonus, this approach is fully
complementary to training-time approaches like DA.

If the foundation models were trained to be invariant to the considered transformations, they couldn't be used to determine which transformation is the canonical. If the downstream model is trained to be invariant, canonicalization is not required.

# Post-rebuttal:
Regarding the compatibility with data augmentation – it may be true that the model becomes more invariant by further adding a canonicalization, but it will not in the balanced case become better on average since the downstream model does not have a preferred orientation if it is trained with data augmentation.

**Essential References Not Discussed:**

Kaba et al. were not the first to propose canonicalization by minimizing an energy score, see for instance

Boominathan, Lokesh, Suraj Srinivas, and R. Venkatesh Babu. "Compensating for large in-plane rotations in natural images." Proceedings of the Tenth Indian Conference on Computer Vision, Graphics and Image Processing. 2016.

**Experimental Designs Or Analyses:**

The paper makes a strong argument for using foundation models for canonicalization, but a large part of the argument is the alleviation of having to train a canonicalization network. In other words, this is an argument that is based on reduced computational requirements. However, since foundation models are quite heavy to run and several transformations have to be tested in a Bayesian optimisation scheme, it seems like the method is heavier than prior work at inference time. This is mentioned in Limitation (1), but the paper should include explicit numbers on compute to give the reader a proper sense of the trade-offs involved.

# Post-rebuttal:
I am satisfied with the addition of computational requirements to the paper.

**Methods And Evaluation Criteria:**

The evaluations make sense.

**Other Comments Or Suggestions:**

N/A

**Other Strengths And Weaknesses:**

The paper is generally well-written and clear.

**Questions For Authors:**

1. What is the computational cost of the method?
2. In Figure 1(b), is the left-most example an output from the model? I.e. does it produce a background-free teddy bear from an image with background?

Including the computational cost in the paper would lead me to raise my score if no critical concerns are shown in other reviews.

# Post-rebuttal:
I am satisfied with the addition of computational requirements to the paper. Thus I will raise my score to Accept as indicated.

**Relation To Broader Scientific Literature:**

The paper shows that the proposed method works. This has not been demonstrated earlier.

**Theoretical Claims:**

N/A

---

> ### Author Rebuttal · Authors · 2025-04-01
>
> Thank you for your insightful feedback. We are glad you found our experimental results to be good and we hope FOCAL can be a key step towards a zero-shot approach towards a wide range of visual invariance. We respond to your concerns and questions below:
>
> - **“If the downstream model is trained to be invariant, canonicalization is not required.”**
>
>     - The difficulty lies in practically achieving a general form of invariance just with data augmentation. Data augmentation still has limitations on long-tailed data [1], generalizing to new transformations, or generalizing outside the trained range. Thus, even if a downstream model is trained with data augmentation, our method can still complement training-time data augmentation by improving invariance in such cases. As an example, in our 2D rotation experiments, FOCAL increases the model’s invariance despite it being trained with data augmentation: even on a balanced CIFAR10 dataset setting, we find that the EKLD with FOCAL+DA is over 10x lower than just DA (0.0053 vs. 0.056).
>       [1] https://arxiv.org/abs/2203.09739
>
> - **“but a large part of the argument is the alleviation of having to train a canonicalization network. In other words, this is an argument that is based on reduced computational requirements … paper should include explicit numbers on compute to give the reader a proper sense of the trade-offs involved.”**
>
>     - We fully agree that it is important to discuss runtime costs, and will add that analysis to the paper. Please see our “How fast does the proposed method run?...” response to XwyL for details.
>     - To clarify, our biggest claimed benefit is not runtime but rather generalization.
>     - Consequently, our method outperforms PRLC’s specialized canonicalizers not only on their trained settings (Figure 7, Table 1) but also on new datasets (e.g, ImageNet) and downstream models (e.g., CLIP). In addition, our approach also works for more complex transformations than previously explored with canonicalization (lighting, 3D, day-night, active vision).
>     - If runtime is a priority and training a canonicalizer is acceptable, our method can still be useful as a source of supervision. Please see our “Distilling FOCAL energy into a cheaper (or one-shot) EBM” response to XwyL.
>
> - **“It contains comparisons to TTA. It is unclear if those comparisons are at an equal compute budget.”**
>
>     - Yes, this comparison uses an equal compute budget.
>
> - **“Boominathan, Lokesh, Suraj Srinivas, and R. Venkatesh Babu. ‘Compensating for large in-plane rotations in natural images.’ Proceedings of the Tenth Indian Conference on Computer Vision, Graphics and Image Processing. 2016.”**
>
>     - Thanks! We agree this is a useful citation and will add it to the paper.
>
> - **“What is the computational cost of the method?”**
>
>     - See “How fast does the proposed method run?...” response to XwyL above
>
> - **“In Figure 1(b), is the left-most example an output from the model? I.e. does it produce a background-free teddy bear from an image with background?”**
>
>     - The left-most example is an output from TRELLIS [2], the 3D generator we used. TRELLIS removes the background as part of its pipeline, as TRELLIS is trained on background removed images.
>       [2] https://arxiv.org/abs/2412.01506

---

### Official Review · Reviewer_tdnV · 2025-03-14

**Overall Recommendation:** 3

**Summary:**

The paper introduces FOCAL, a zero-shot framework designed to achieve invariant perception at test-time using pre-trained foundation models like CLIP and Stable Diffusion. FOCAL generates transformed versions of input images and selects a canonical version by minimizing an energy function derived from these models, requiring no additional training or architectural changes.

**Claims And Evidence:**

The authors claim that vision foundation models, such as CLIP, perform poorly on direct classification tasks when faced with viewpoint, color, or lighting changes. But they exhibit strong performance in ranking (**the ability to select the most plausible (canonical) image from among multiple candidate images** via energy-based forms) multiple transformed images by their probability of belonging to the dataset distribution. They propose leveraging this capability to canonicalize images, improving classification accuracy.

However, the paper lacks an analytical explanation regarding why foundation models, despite their reduced direct classification accuracy under transformations, possess robust and accurate ranking abilities, even without finetuning on downstream tasks. For instance, Figure 5 suggests CLIP inherently identifies canonical viewpoints even without task-specific fine-tuning, outperforming a fine-tuned version in terms of viewpoint robustness. The paper provides strong experimental results but does not sufficiently explore or explain the underlying mechanisms or reasons behind this key capability.

**Essential References Not Discussed:**

None.

**Experimental Designs Or Analyses:**

What is the inference time of the proposed method compared to baselines such as finetuned CLIP, TTA, and PRLC?

**Methods And Evaluation Criteria:**

As acknowledged by the authors, estimating the canonical form for every image involves multiple transformations, necessitating the use of Gaussian processes for efficient search. Despite using GP, the proposed approach still likely demands significantly more computational resources. Furthermore, in realistic scenarios, images typically undergo multiple combined transformations rather than isolated ones (e.g., viewpoint + color + contrast + active vision). While evaluating single transformations separately is understandable for methodological clarity, this raises concerns that employing Bayesian optimization to search through a more complex and realistic transformation space could become increasingly infeasible.

Additionally, Figure 5 shows that finetuned CLIP performs better for samples with high viewpoint rankings (easy samples). How can we determine, given a specific sample image, whether naive finetuned CLIP or FoCAL would yield better performance?

**Other Comments Or Suggestions:**

None.

**Other Strengths And Weaknesses:**

None.

**Questions For Authors:**

In section B.3, the weighting between the diffusion and CLIP energy terms is currently set via hyperparameters. Could the authors clarify under which circumstances one energy term (diffusion or CLIP) is more critical than the other, and what specific roles each model plays in different canonicalization scenarios?

**Relation To Broader Scientific Literature:**

Proposed model could be used in the future to enhance the test-time performance of foundation models.

**Theoretical Claims:**

There are no theoretical claims to discuss.

---

> ### Author Rebuttal · Authors · 2025-04-01
>
> Thank you for your insightful feedback. We are glad you recognize our empirical strengths in canonicalizing images on a range of transformations, and we hope FOCAL can be a key step towards a zero-shot approach. We respond to your concerns and questions below:
>
> - **“The paper lacks an analytical explanation regarding why foundation models, despite their reduced direct classification accuracy under transformations, possess robust and accurate ranking abilities, even without finetuning on downstream tasks.”**
>
>      - That’s a good point, and we will add a more detailed explanation about this in section 4 as outlined below.
>
>      - Specifically, we assume: (1) there is at least one in-distribution image in the set of transformed images, and (2) the foundation models can be used as a prior where ID images have lower energy than OOD. This has been shown/used previously by [1, 2].
>
>        [1] https://arxiv.org/abs/2206.09012
>
>        [2] https://arxiv.org/abs/1912.03263
>
>     - If these two assumptions hold for a given sample, the energy minimization scheme returns an image that is in-distribution. Importantly, this scheme only requires the foundation models to distinguish between ID vs OOD images. Even if the classification and energy values for OOD images are not well-behaved, our scheme still works as long as in-distribution images have lower energy than out-of-distribution images.
>
>        We acknowledge that the limits of foundation models to serve as image priors are still not fully understood. There may be transformations or OOD images for which assumption 2 may not hold. A rigorous understanding of exactly when foundation models work well as image priors is a great future direction.
>
>
>
> - **“While evaluating single transformations separately is understandable for methodological clarity, this raises concerns that employing Bayesian optimization to search through a more complex and realistic transformation space could become increasingly infeasible.”**
>
>     - This is a fair question. In general, handling multiple and complex transformations remains a difficult open problem. For our approach, we agree that more complex transformations (resulting in a higher-dimensional optimization problem) would likely lead to higher sample complexity, which would make the optimization more difficult.
>
>     - However, Bayesian optimization and gradient descent for latent space optimization have been successful in other mid-to-high dimensional optimization problems like protein design [3,4,5,6]. We are still figuring out how to best use these techniques for invariance, and we think it could be a great future research direction. We hope our paper serves as a strong foundation to explore this direction and hope to solve it in future work.
>       [3]: https://arxiv.org/abs/2006.09191
>       [4]: https://arxiv.org/abs/2201.11872
>
>       [5]: https://arxiv.org/abs/1610.02415
>
>       [6]: https://www.nature.com/articles/s42256-022-00532-1
>
> - **“Additionally, Figure 5 shows that finetuned CLIP performs better for samples with high viewpoint rankings (easy samples). How can we determine, given a specific sample image, whether naive finetuned CLIP or FoCAL would yield better performance?”**
>
>     - Great question! Usually, the more in-distribution/upright an image is, the more likely it is that finetuned CLIP will do better. As a heuristic: if the given image is in a local minimum of the energy function, FOCAL will likely not be helpful. We used this to create a decision rule for 2D rotations that allows us to skip canonicalization where it is unnecessary with 95% accuracy (see “only using canonicalization when necessary” in our response to Xwyl above). We will add these results to the appendix.
>
> - **“Could the authors clarify under which circumstances one energy term (diffusion or CLIP) is more critical than the other, and what specific roles each model plays in different canonicalization scenarios?”**
>
>     - Also a great question. We don’t have a general theoretical answer, but we have run some ablations (e.g., Table 4 in the appendix). In our experience, CLIP is more important when there is a clearly defined object in the image with an easily defined caption (e.g., Objaverse classification). Diffusion helps much more for scenes with many objects (e.g., in segmentation) and visual structures (e.g., edges). We currently find the best combination via hyperparameter optimization, but hope to have a more general approach in future work.

---

> > ### Comment · Reviewer_tdnV · 2025-04-03
> >
> > Thanks for the authors' rebuttal. I will keep the positive score.

---

### Official Review · Reviewer_XwyL · 2025-03-16

**Overall Recommendation:** 4

**Summary:**

This paper introduces Foundation-model guided Canonicalization (FOCAL), a novel zero-shot framework designed to enhance the invariance of vision models to various transformations by test-time optimization. The method generates candidate transformations (e.g., 3D novel views) and selects a canonical version by optimizing an energy function derived from the foundation models like CLIP and Stable Diffusion. The authors demonstrate the effectiveness of FOCAL across 2D/3D rotations, illumination shifts, and day-night changes, showing improved robustness for models like CLIP and SAM without requiring any training or architectural modifications. The work also explores applications in active vision.

**Claims And Evidence:**

- FOCAL improves robustness to various transformations in a zero-shot manner.
	- The experiments presented in Section 4 provide empirical support for this claim, where they show improved performance on viewpoint shifts, illumination changes, 2D rotations, and day-night transformations.
- FOCAL outperforms or matches task-specific canonicalizers like PRLC in their trained settings, despite being zero-shot.
	- The 2D rotation experiments in Section 4.3 and Figure 7 support this claim. FOCAL matches or surpasses PRLC's performance on 2D rotation tasks and demonstrates better generalization on ImageNet.
- The proposed approach is fully complementary to training-time approaches like DA.
	- To make this claim, don’t they need to show that FOCAL + DA outperforms FOCAL alone? But I couldn’t find any such experiment.

**Essential References Not Discussed:**

- No major concerns here

**Experimental Designs Or Analyses:**

- The experiments described in Section 4 are comprehensive (e.g viewpoint invariance, Color and Contrast, 2D rotation, day-night transformation, active vision), and for each task, they use multiple datasets to verify the results.
- Hyperparameters for the energy functions are explained in Section B.3.

**Methods And Evaluation Criteria:**

- The proposed method is well-motivated and clearly explained. The "vary and rank" scheme is intuitive. The energy functions derived from CLIP and Stable Diffusion seem appropriate.
- The evaluation criteria are suitable for assessing the method's effectiveness. The choice of datasets (ImageNet, CIFAR, Objaverse-LVIS, CO3D) and transformations (viewpoint shifts, illumination changes, 2D rotations, day-night) is comprehensive and relevant.

**Other Comments Or Suggestions:**

- No major typos.

**Other Strengths And Weaknesses:**

- I think their idea of using foundation models to optimize the energy function for guiding canonicalization is clever, and the experiments demonstrate its strong performance compared to previous approaches.

**Questions For Authors:**

- How fast does the proposed method run? The paper states that "evaluating the energy function for many candidates is computationally expensive," but I think providing a more concrete runtime would be more helpful for the audience.
- On a related note, could you elaborate on potential strategies for improving the computational efficiency of FOCAL?

**Relation To Broader Scientific Literature:**

- The authors clearly discuss the limitations of data augmentation and equivariant networks, highlighting the challenges they face in open-ended scenarios.
- The paper builds upon the theoretical foundations laid by Kaba et al. (2022) and leverages the energy-based model perspective from Grathwohl et al. (2019) and the use of diffusion models as priors from Graikos et al. (2022).

**Theoretical Claims:**

- No proof provided.

---

> ### Author Rebuttal · Authors · 2025-04-01
>
> Thank you for your thoughtful review. We are glad you found our use of foundation models for optimizing the energy function clever and we hope FOCAL can be a key step towards a zero-shot approach towards a wide range of visual invariance. We address your concerns and questions below:
>
> * **“‘The proposed approach is fully complementary to training-time approaches like DA.’ To make this claim, don’t they need to show that FOCAL + DA outperforms FOCAL alone? But I couldn’t find any such experiment.”**
>
>    - Good point. We find that FOCAL+DA indeed outperforms either FOCAL or DA by themselves, and will add these results to the main paper. As an example, we find that on a ResNet32 trained on CIFAR10, the C8 rotated accuracy is 72% with just DA, 72.1% with just FOCAL, and 73.2% with FOCAL+DA.
>
> * **“How fast does the proposed method run? The paper states that ‘evaluating the energy function for many candidates is computationally expensive,’ but I think providing a more concrete runtime would be more helpful for the audience.”**
>
>     - Good question. Our paper focuses on generalization rather than efficiency, but we agree that computational cost is important, and will include it in the paper. Our method’s cost is:
>
>        (# of transforms evaluated) X (Cost of transforming + Cost of evaluating [CLIP + Diffusion model] + Cost of inference).
>
>        **Example**: Consider 2D rotation (with 8 rotations around the circle). When using CLIP as the downstream model and 5 diffusion steps, we must evaluate the energy for each of the 8 rotations. Thus, our method requires roughly 56x more FLOPs compared to standard inferencing. As for latency, our procedure is highly parallelizable, and the latency can be close to standard inference latency.
>
>          As another point of comparison, consider a test-time-augmentation (TTA) strategy that just averages the outputs. Such a strategy would use 8x FLOPs than the naive classifier. Thus, in comparison to TTA, our approach is 7x more expensive in FLOPs.
>
>          We will add a more detailed runtime analysis in the camera-ready paper, and a table detailing the FLOPs and runtime for each experiment (in seconds).
>
> * **“Could you elaborate on potential strategies for improving the computational efficiency of FOCAL?”**
>
>      - **Only using canonicalization when necessary**: Similar to the “mental rotation” [1] phenomenon in humans, where we classify familiar poses quickly but go through a slow mental uprighting process to classify unfamiliar poses, one approach is to apply canonicalization selectively.
>
>          We have preliminary results for an approach that may significantly reduce amortized computational complexity by skipping the canonicalization when unnecessary.
>
>          For example, for 2D rotations, we can skip the canonicalization by comparing the image’s CLIP energy against +90 and -90 degree rotations, and thresholding the energy difference. We can use this simple classifier to detect upright vs. non-upright with 95% accuracy.
>          In summary, with only 3 CLIP inferences, we can detect whether to do canonicalization with 95% accuracy. If the extreme rotations are rare (as is typical in the real world), this can bring significant computational savings and, on average, be even cheaper than TTA.
>
>           [1] https://psycnet.apa.org/record/1971-28060-001
>
>
>      - **Fewer diffusion steps**: The majority of our computational cost comes from the diffusion energy, especially because it uses multiple steps. Diffusion classifier [2] has done an extensive analysis of efficient diffusion step schedules that can be used to rank inputs (in the context of classification). We didn’t explore these schedules since our focus was to show that CLIP and SD can be used for invariance to diverse and complex transformations, but leveraging efficient diffusion step schedules can be a promising future direction.
>
>          [2] https://arxiv.org/pdf/2303.16203
>
>      - **Distilling FOCAL energy into a cheaper (or one-shot) EBM**: Since the desired output is only a scalar energy value, it might be possible to distill CLIP and SD’s energy function into a smaller single-shot EBM like [3]. This is yet another promising future research direction.
>
>           [3]: https://implicit-pdf.github.io/

---

### Decision · Program_Chairs · 2025-05-01

**Decision:**

Accept (poster)

**Comment:**

The paper initially received ratings of (4, 3, 2, 3) from reviewers. After the rebuttal, the ratings were raised to (4, 3, 4, 4), all of which are positive.

Reviewers noted that "the paper is well-written and easy to follow." They also found the proposed idea "of using foundation models to optimize the energy function for guiding canonicalization is clever, and the experiments demonstrate its strong performance compared to previous approaches." Reviewers expressed concerns about the computational issues of the proposed method, but the rebuttal effectively addressed these concerns.

As a result, the area chair agrees with the reviewers' assessments and recommends the acceptance of the paper. Furthermore, the area chair encourages the authors to revise the paper as promised, in particular, to add "a more detailed runtime analysis in the camera-ready paper, and a table detailing the FLOPs and runtime for each experiment (in seconds)."